# The centrosomal protein 131 participates in the regulation of mitochondrial apoptosis

Clotilde C. N. Renaud[1,2], Kilian Trillet[1,2], Jane Jardine[1,2], Laura Merlet[1,2], Ophélie Renoult[3],
Mélanie Laurent—Blond [3], Zoé Catinaud[1,2], Claire Pecqueur [3], Julie Gavard [1,2,4] & Nicolas Bidère [1,2✉]

Centriolar satellites are multiprotein aggregates that orbit the centrosome and govern centrosome homeostasis and primary cilia formation. In contrast to the scaffold PCM1, which nucleates centriolar satellites and has been linked to microtubule dynamics, autophagy, and intracellular trafficking, the functions of its interactant CEP131 beyond ciliogenesis remain unclear. Using a knockout strategy in a non-ciliary T-cell line, we report that, although dispensable for centriolar satellite assembly, CEP131 participates in optimal tubulin glycylation and polyglutamylation, and microtubule regrowth. Our unsupervised label-free proteomic analysis by quantitative mass spectrometry further uncovered mitochondrial and apoptotic signatures. CEP131-deficient cells showed an elongated mitochondrial network. Upon cell death inducers targeting mitochondria, knockout cells displayed delayed cytochrome c release from mitochondria, subsequent caspase activation, and apoptosis. This mitochondrial permeabilization defect was intrinsic, and replicable in vitro with isolated organelles. These findings extend CEP131 functions to life-and-death decisions and propose ways to interfere with mitochondrial apoptosis.

[1] Team SOAP, CRCI2NA, Nantes University, INSERM, CNRS, Université d'Angers, Nantes, France. [2] Equipe Labellisée Ligue Contre le Cancer, Nantes, France. [3] Team PETRY, CRCI2NA, Nantes University, INSERM, CNRS, Université d'Angers, Nantes, France. [4] Institut de Cancérologie de l'Ouest (ICO), Saint-Herblain, France. ✉email: nicolas.bidere@inserm.fr

Centriolar satellites are small non-membranous proteinaceous particles of 70–100 nm that crawl along microtubules in a motor-dependent manner and gravitate toward the centrosome[1]. These high-order assemblies are remarkably dynamic and they quickly disassemble during mitosis to reappear over interphase[2–4]. They also acutely redistribute in the cytosol through a p38-MK2 axis in response to external cues such as UV radiation or heat shock[5,6]. Centriolar satellites ensure protein cargoes trafficking to the centrosome, thereby regulating its homeostasis. They also participate in the formation of the primary cilia, a microtubule-based structure that projects out from the cell surface to act as a "sensing antenna"[7,8]. Genetic perturbation affecting centriolar satellite encoding genes can jeopardize ciliogenesis and cause human ciliopathies[9,10]. Nearly 600 proteins have been identified in the interactome of 22 centriolar satellite components, among which the scaffold PCM1 is the most characterized[11,12]. PCM1 nucleates the satellites, regulates the abundance of their components, prevents their relocation at the centrosome, and allows optimal ciliogenesis[7,9,10,13–15]. Nevertheless, growing evidence also suggests that PCM1 and centriolar satellites participate in additional functions. These include the maintenance of global proteostasis via autophagy and the ubiquitin-proteasome system, endosomal trafficking, neurogenesis, as well as the regulation of post-translational modifications of the tubulin by controlling the assembly and the stability of the tubulin polyglutamylase complex (TPGC)[9,10,16–19].

The Centrosomal Protein 131 (CEP131, also called AZI1) has been initially identified as a centrosomal protein by proteomic analysis and immunofluorescence in ectopic conditions[20]. It was subsequently shown that endogenous CEP131 also localizes to the core centriolar region and is part of centriolar satellites in interphase[9,21–24]. Accordingly, CEP131 binds to PCM1 and is found in the interactome and the proximitome of known centriolar satellite elements[9,11,12,21]. Further illustrating the shared distribution of CEP131 between centrosome and satellites, CEP131 accumulates at the centrosome in cells knockout for PCM1 or in mitotic cells when satellites are disassembled[9,21]. In contrast to PCM1, the loss of CEP131 does not cause a drastic redistribution of centriolar satellites[21,23]. Besides suppressing the formation of primary cilia when acutely depleted[25], CEP131 has been linked with the clearance of aggresomes, centrosome amplification, cell proliferation, and genomic stability maintenance[18,21,26]. Nevertheless, the exact landscape of its functions remains elusive. To gain further insights into the roles of CEP131, we ablated its expression in a lymphoblastoid T cell line, which cannot assemble primary cilia[27]. Here, we report that CEP131 depletion affects the global proteome of the cells, and allows optimal antigen receptor signaling and mitochondrial apoptosis.

## Results and discussion

### CEP131 is dispensable for centriolar satellite organization and participates in microtubule dynamics.
To assess the role of CEP131 beyond ciliogenesis, we undertook a knockout strategy by CRISPR/Cas9 gene editing technology in the Jurkat T cell line, which fails to form primary cilia. Two clones, namely #1 and #2, with different bi-allelic mutations in the CEP131 gene, as assessed by genomic DNA sequencing, were selected (Fig. 1a). As expected, CEP131 was not expressed at the mRNA and protein level (Fig. 1b, c). The deletion or amplification of CEP131 has been linked to mitotic and centrosome abnormalities, proliferation defects, and signs of DNA damage dependent on the cell type[21,24,28]. In CEP131 knockout Jurkat cells, the analysis of cell cycle only showed modest changes although a decrease in cell proliferation was observed, supporting previous works[21,23]

(Supplementary Fig. 1a, b). Moreover, no drastic change in the cell cycle was observed when CEP131 was silenced in the mouse fibroblast cell line L929 (Supplementary Fig. 1c). We further checked for hallmarks of chromosomal instability and did not observe increased numbers of micronuclei in CEP131 knockout Jurkat cells. Moreover, no difference was observed in cells treated with the MPS1 inhibitor Reversine, a compound known to induce micronuclei[29] (Supplementary Fig. 1d). Furthermore, the phosphorylation of Chk2 and ATM, which reflects DNA damage and repair, remained undetectable in CEP131 knockout Jurkat (Supplementary Fig. 1e). As expected[21,23], the deletion of CEP131 did not overtly change the abundance of several centriolar satellite components, including PCM1, CEP290, MIB1, CEP72, CCDC66, OFD1, and BBS4 (Fig. 1d). Moreover, we observed that PCM1, which scaffolds centriolar satellites[1], clustered around the centrosome regardless of CEP131 (Fig. 1e). Accordingly, the measurement of the PCM1 intensity using radial profile analysis was not affected in CEP131 knockout cells (Fig. 1f, g). Co-immunoprecipitation experiments further showed that PCM1 efficiently interacted with the centriolar satellite components CEP290, CEP72, and MIB1 in the absence of CEP131 (Fig. 1h). We next wondered if CEP131 participates in some functions recently ascribed to PCM1 and centriolar satellites, including the regulation of microtubule nucleation and regrowth and of post-translational modifications of tubulin that determine microtubule dynamics and microtubule-associated proteins (MAPs) interactions[19,30]. First, we observed a reduction in the regrowth of microtubules after depolymerization induced by nocodazole treatment in the absence of CEP131 (Fig. 1i). Moreover, we also noticed a decrease in the abundance of glycylated and polyglutamylated species of tubulin but not of detyrosination modifications in CEP131-deleted cells, suggesting a change in the tubulin code (Supplementary Fig. 1f, g).

Taken together, our data suggest that CEP131 is dispensable for the general distribution and organization of centriolar satellites and may participate in microtubule dynamics.

### CEP131 regulates the abundance of proteins involved in life-and-death decisions.
To gain further insights into the overall impact of CEP131 depletion, we next conducted an unsupervised proteomic analysis of WT and CEP131 knockout Jurkat cells by label-free quantitative proteomics. Among the 7285 proteins identified with high confidence, 542 were downregulated and 309 were upregulated in CEP131[KO#1] cells ($p_{value} < 0.05$, $|\log_2(FC)| > 0.5$) (Fig. 2a). Of note, no overt changes in the abundance of the tested centriolar satellite components was observed, further validating the proteomic analysis (Fig. 2b). A comparison of our proteomic dataset with centrosomal proteins (GO Cellular Component: GO:0005813) showed that the deletion of CEP131 had only a modest impact on the abundance of centrosomal proteins (Fig. 2c). This is in line with the recent analysis, by Odabasi and coworkers, of the proteome of mouse kidney epithelial IMCD3 cells knockout for PCM1[10]. Moreover, 63 differentially regulated proteins were shared between this previous work and our current analysis of CEP131 knockout Jurkat cells, among which 23 were centrosomal proteins (Fig. 2d). Hence, our results suggest that CEP131, similarly to PCM1, only marginally affects the abundance of centrosomal proteins.

The KEGG enrichment analysis of downregulated proteins in CEP131 knockout cells identified a signature related to the T cell receptor (TCR) signaling, with a reduced abundance of several proximal regulators of the pathway (Fig. 2e, Supplementary Fig. 2a, b). Accordingly, the expression of some of the proteins forming the TCR such as TCRαβ and CD3ζ was decreased, as assessed by immunoblotting and/or flow cytometry analyses

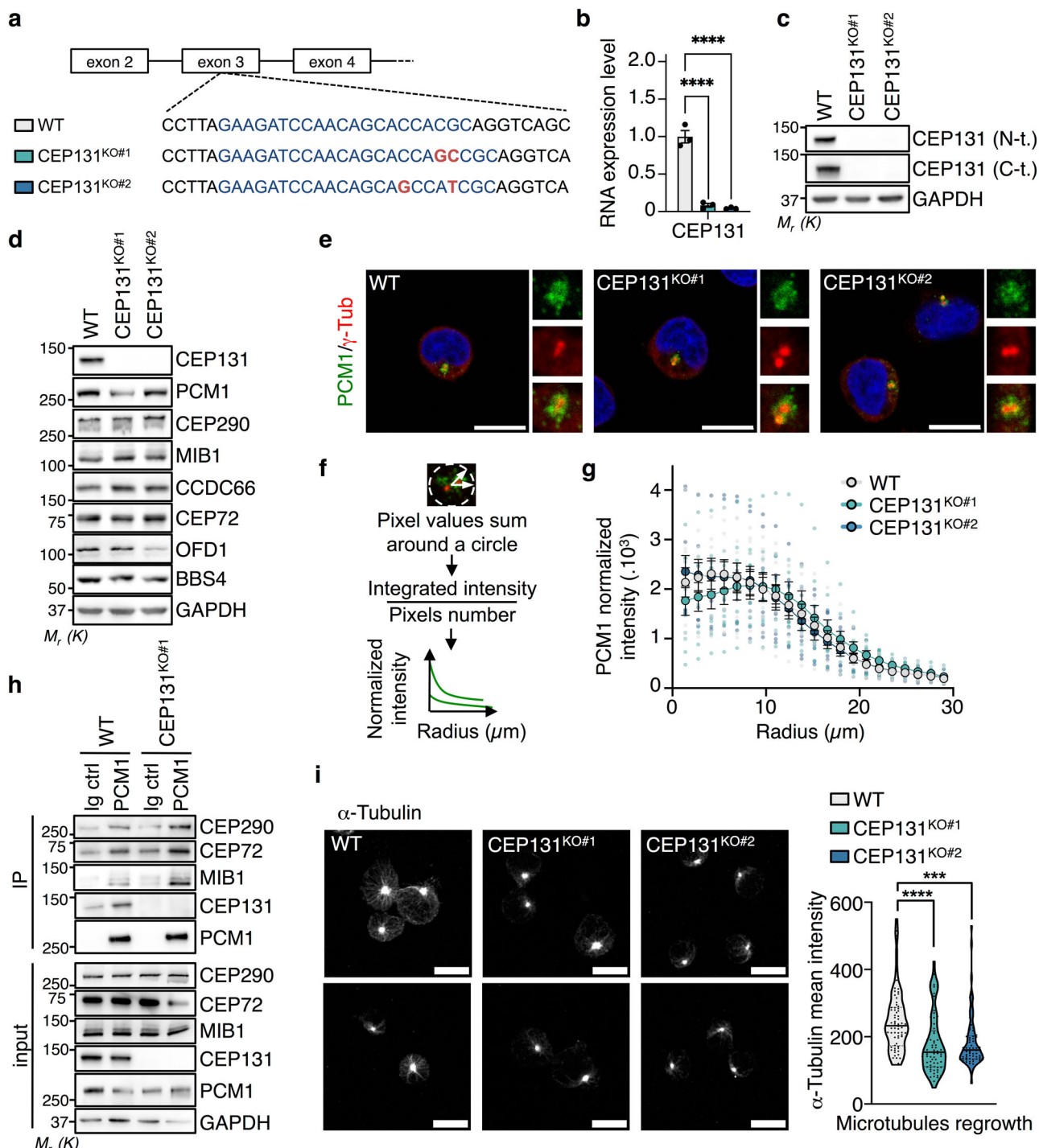

**Fig. 1 Characterization of CEP131 knockout Jurkat cells. a** Schematic representation of the CEP131 gene, with positioning of the CRISPR sequence guide (blue) in exon 3. Genomic sequences in wild-type (WT) and two bi-allelic clones (KO#1 and KO#2; mutations in red) are shown. **b** The RNA expression level of CEP131 was assessed by qPCR in WT, CEP131KO#1, and KO#2 cells (mean ± SEM, $n = 3$ biological replicates, fold change using ACTB and HPRT1 as housekeeping genes for normalization, one-way ANOVA, ****$p < 0.0001$). **c, d** Lysates from WT, CEP131 KO#1, and KO#2 Jurkat cells were prepared and analyzed by immunoblotting with antibodies specific to the indicated proteins. **e** Representative immunofluorescence images of PCM1 (green) and γ-tubulin (red) in WT, CEP131 KO#1, and KO#2 Jurkat cells. Nuclei were counterstained with 4'−6-diamidino-2- phenylindole (DAPI). Scale bars, 10 μm. **f, g** Radial profile analysis of PCM1 in a 29 μm circle whose center is defined by γ-tubulin, in WT, CEP131 KO#1, and KO#2 Jurkat cells (mean ± SEM of $n = 10$ cells/condition). **h** Cell lysates from WT, CEP131 KO#1, and KO#2 Jurkat were immunoprecipitated (IP) with antibodies against PCM1 or with non-relevant Ig (negative control). Samples were then analyzed by immunoblotting as indicated. Inputs are the total lysates collected before the IP. **i** WT, CEP131 KO#1, and KO #2 Jurkat cells were treated with 10 μM nocodazole for 1 h, prior to washing. Microtubule regrowth was assessed after 5 min by confocal microscopy analysis of α-tubulin. Representative images of three independent experiments (left) (scale bars, 10 μm), and quantification of α-tubulin intensity of one representative experiment (right) ($n = 70$ cells/condition; one-way ANOVA, ****$p < 0.0001$). Data information: (**c, d, h**) GAPDH served as a loading control. Molecular weight markers ($M_r$) are shown. Data are representative of three independent experiments.

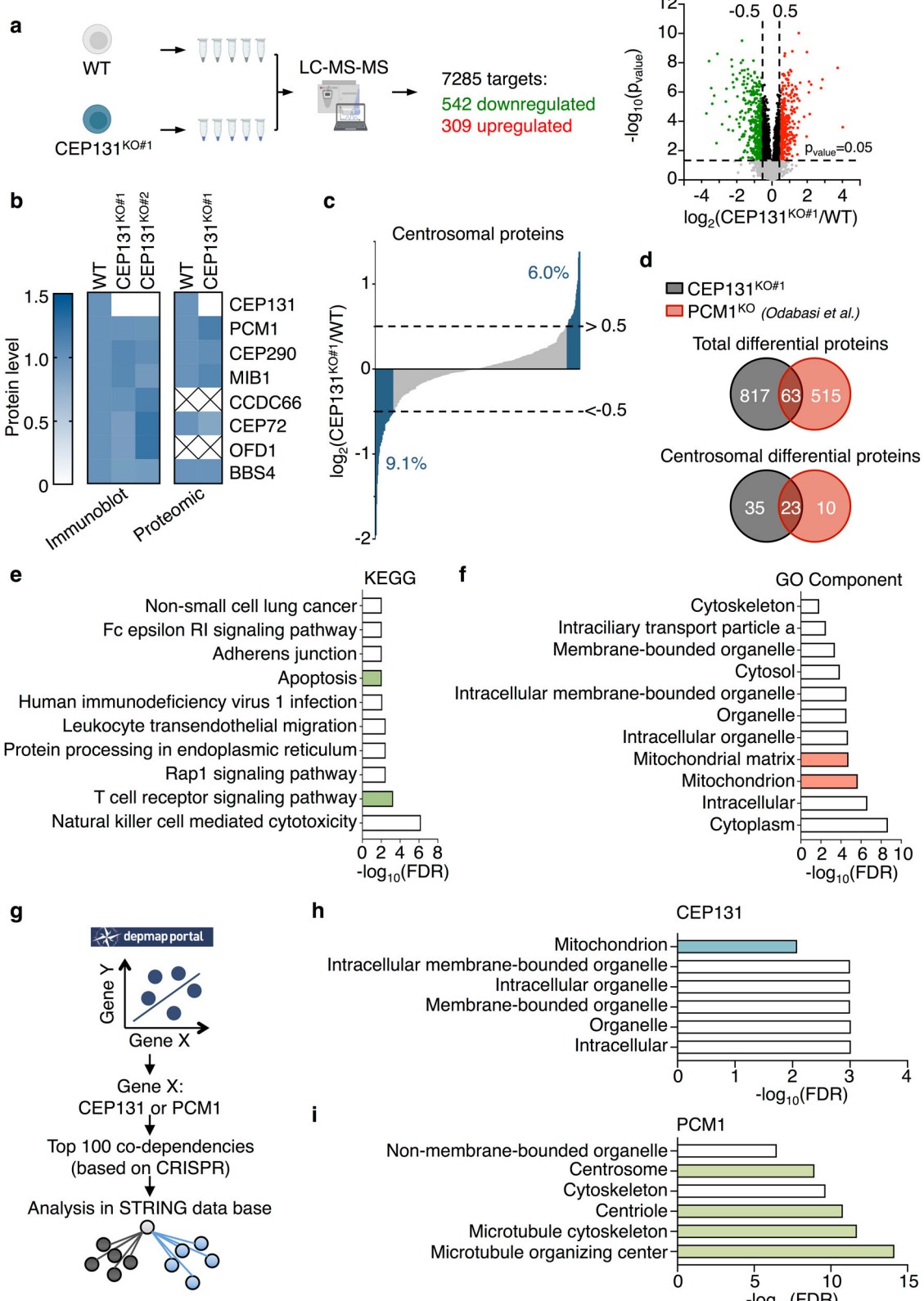

(Supplementary Fig. 2c, d). This was however not the case at the RNA level, indicating a post-transcriptional regulation (Supplementary Fig. 2e). Of note, the total level of CD3ε was not overtly affected without CEP131, but less was detected at the cell surface, suggesting a defect in trafficking to the membrane and/or its stability at the cell surface (Supplementary Fig. 2c–f).

To test this hypothesis, cells were treated with Phorbol 12,13-dibutyrate (PDBu) to force TCR internalization and recycling[31]. We observed that the surface recovery of CD3ε was reduced in CEP131 knockout cells, as assessed by flow cytometry analysis (Supplementary Fig. 2g). The ligation of antigen receptors initiates intracellular signaling cascades culminating in the

**Fig. 2 CEP131 knockout Jurkat cells display an apoptosis and mitochondrial signature. a** Schematic overview of the mass spectrometry analysis performed on five biological replicates of wild-type (WT) or CEP131 KO#1 Jurkat cells. The volcano plot illustrates the 7285 detected proteins of which 542 are downregulated and 309 are upregulated. Proteins were classified as downregulated or upregulated if the $p_{value} < 0.05$ and $|\log_2(\text{fold change})| > 0.5$. **b** Heatmaps representing the abundance of centriolar satellite components based on the densitometric analysis performed on immunoblotting of lysates from WT, CEP131KO#1, and KO#2 Jurkat cells (mean of $n = 3$ biological replicates) and on the proteomic analysis of WT and CEP131 KO#1 Jurkat cells (mean of $n = 5$ biological replicates). **c** Comparison of the CEP131 KO#1 Jurkat cells proteome to the centrosomal proteome (GO:0005813). Upregulated and downregulated proteins among the 385 detected centrosomal ones are shown in blue and percentages are indicated ($|\log_2(\text{FC})| > 0.5$). **d** Comparison of the proteome of CEP131 KO#1 Jurkat cells with the one of PCM1 knockout IMCD3 cells[10]. The top Venn diagram shows total differential proteins and the bottom Venn diagram is focused on centrosomal differential proteins ($p_{value} < 0.05$, $|\log_2(\text{FC})| > 0.5$). **e** The top 10 KEGG pathways enriched for the proteins significantly downregulated in CEP131 KO#1 Jurkat cells ($p_{value} < 0.05$, $\log_2(\text{FC}) < -0.5$). **f** The top 10 GO Component enrichment for the proteins significantly upregulated in CEP131 KO#1 Jurkat cells is presented ($p_{value} < 0.05$, $\log_2(\text{FC}) > 0.5$). **g–i** Workflow of the data mining performed (**g**). The top 100 genes that presented co-dependencies (based on CRISPR screening) with CEP131 or PCM1 were extracted from the depmap portal database. Analysis of the enrichment was done on the STRING database and compartments enriched in the top 100 co-dependencies with CEP131 (**h**) or PCM1 (**i**) are presented. Data information: (**e**, **f**, **h**, **i**) data are represented using the false discovery rate (FDR).

activation of transcription factors including NF-κB[32]. We thus monitored the impact of CEP131 deletion on the transcription of 84 NF-κB target genes upon stimulation with antibodies to CD3 and CD28 (Supplementary Table 2). This showed an overall decreased induction of human NF-κB target gene panel (Supplementary Fig. 2h). In keeping with this, total tyrosine phosphorylation was reduced upon CD3 cross-linking in both CEP131 knockout clones (Supplementary Fig. 2i). The same was true for the phosphorylation of the key TCR signaling cascade components PLCγ1, ERK, IκBα, and the NF-κB subunit p65, in cells stimulated with anti-CD3 and anti-CD28 (Supplementary Fig. 2j). By contrast, signaling normally occurred when the TCR activation is bypassed by using PMA plus ionomycin (Supplementary Fig. 2j). These results suggest that CEP131 participates in the maintenance of the abundance of proximal TCR components and in the regulation of CD3ε trafficking.

The KEGG enrichment analysis of downregulated proteins also identified pathways related to "apoptosis" (Fig. 2e). In parallel, the enrichment analysis of gene ontology: component (GO: Component) focused on upregulated proteins unveiled a mitochondrial signature (Fig. 2f). Moreover, we searched for coordinated dependencies with CEP131 across the Cancer Dependency Map Depmap portal (https://depmap.org/portal/), a database that regroups results from genome-wide CRISPR/Cas9 knockout screens in hundreds of cancer cell lines (Fig. 2g). The analysis of the top-hundred genes that display co-dependencies with CEP131 also showed a mitochondrial signature (Fig. 2h). However, PCM1 was correlated with a signature related to microtubules and the centrosome (Fig. 2i). Taken together, these data suggest that CEP131 depletion establishes an apoptosis and mitochondrial signature.

**CEP131 allows optimal mitochondria-dependent cell death.** We next explored further the potential link between CEP131 and mitochondria. First, our evaluation of the mitochondrial mass by flow cytometry using MitoTracker Green showed no overt differences between control and CEP131 knockout cells (Fig. 3a). Accordingly, the ratio between mitochondrial and nuclear DNA, as measured by qPCR, was unchanged without CEP131 (Fig. 3b). The assessment of mitochondrial membrane potential by flow cytometry with tetramethylrhodamine methyl ester (TMRM) unveiled a significant increase in CEP131 knockout cells (Fig. 3c). Using the Seahorse technology, we noted an alteration of the respiratory capacities of the cells in one of the two CEP131 knockout cell lines, suggesting a clonal effect (Supplementary Fig. 3a–c). To next investigate the impact of CEP131 on mitochondrial dynamics, we analyzed the mitochondrial resident protein TOM20 by structure illuminating microscopy (SIM). This unveiled larger segmented 3D TOM20-positive structures in the absence of CEP131, suggesting a more interconnected

mitochondrial network (Fig. 3d, e). Accordingly, a higher number of voluminous mitochondria was observed in CEP131 knockout cells (Fig. 3f). Proper mitochondrial dynamics has been linked to the removal of damaged or superfluous mitochondria by autophagy, a process named mitophagy[33]. To test the impact of CEP131, cells were therefore treated with the electron transport chain inhibitors oligomycin and antimycin, to depolarize mitochondria and thereby trigger mitophagy[34]. As expected, mitochondria depolarization led to the processing of OPA1, the accumulation of the kinase PINK1, which activates the subsequent ubiquitination of outer mitochondrial membrane substrates such as MFN1 and phosphorylates nearby ubiquitin moieties in control cells. Yet, the hallmarks of mitophagy were not changed in CEP131 knockout cells, suggesting that CEP131 is dispensable for the execution of mitophagy (Fig. 3g). Collectively, these data suggest that CEP131 participates in mitochondrial dynamics.

Mitochondrial fusion/fission plays a role in timely regulating programmed cell death by apoptosis. For instance, mitochondria fission facilitates the permeabilization of the mitochondrial outer membrane during apoptosis[35–39]. This notably enables the release of cytochrome c, a key step in the activation of caspases (CASPs) and programmed cell death by apoptosis[40]. To investigate whether CEP131 deletion limits mitochondrial apoptosis, we first measured the drop of the potential of the mitochondrial membrane, an early hallmark of cell death associated with cytochrome c release and CASP activation[41]. We found that the decrease in TMRM staining, which reflects mitochondrial potential dissipation, was delayed in CEP131 knockout cells exposed to the mitochondrial apoptosis inducer Raptinal (Fig. 4a). To further examine the initial burst of cytochrome c release, cells were treated with Raptinal in the presence of the pan-CASP inhibitor z-VAD, to prevent additional CASP activation loop. This showed that the release of cytochrome c from mitochondria was delayed in the absence of CEP131 (Fig. 4b). As a consequence, the subsequent cleavage of CASP3 and CASP8, as well as of the CASP substrates PARP, CYLD, RIPK1, and of proluminescent CASP3/7 and CASP8 substrates was hindered (Fig. 4c, d). Similar results were obtained in cells treated with TNFα in the presence of the Smac-mimetic Birinapant to drive apoptosis[42] (Supplementary Fig. 3d, e). By contrast, the ligation of the death receptor Fas led to similar activation of CASPs in WT and CEP131 knockout cells (Supplementary Fig. 3f, g). From a functional standpoint, we found that cell death was likewise reduced in CEP131 knockout cells exposed to stimuli that involved the mitochondria, such as Raptinal, inhibitors of the BCL-2 family members (ABT-199, S63845, A1155463), staurosporine, etoposide, and TNFα (Fig. 4e and Supplementary Fig. 3h, i). Consistent with the biochemistry results, Fas crosslinking killed cells regardless of CEP131 expression

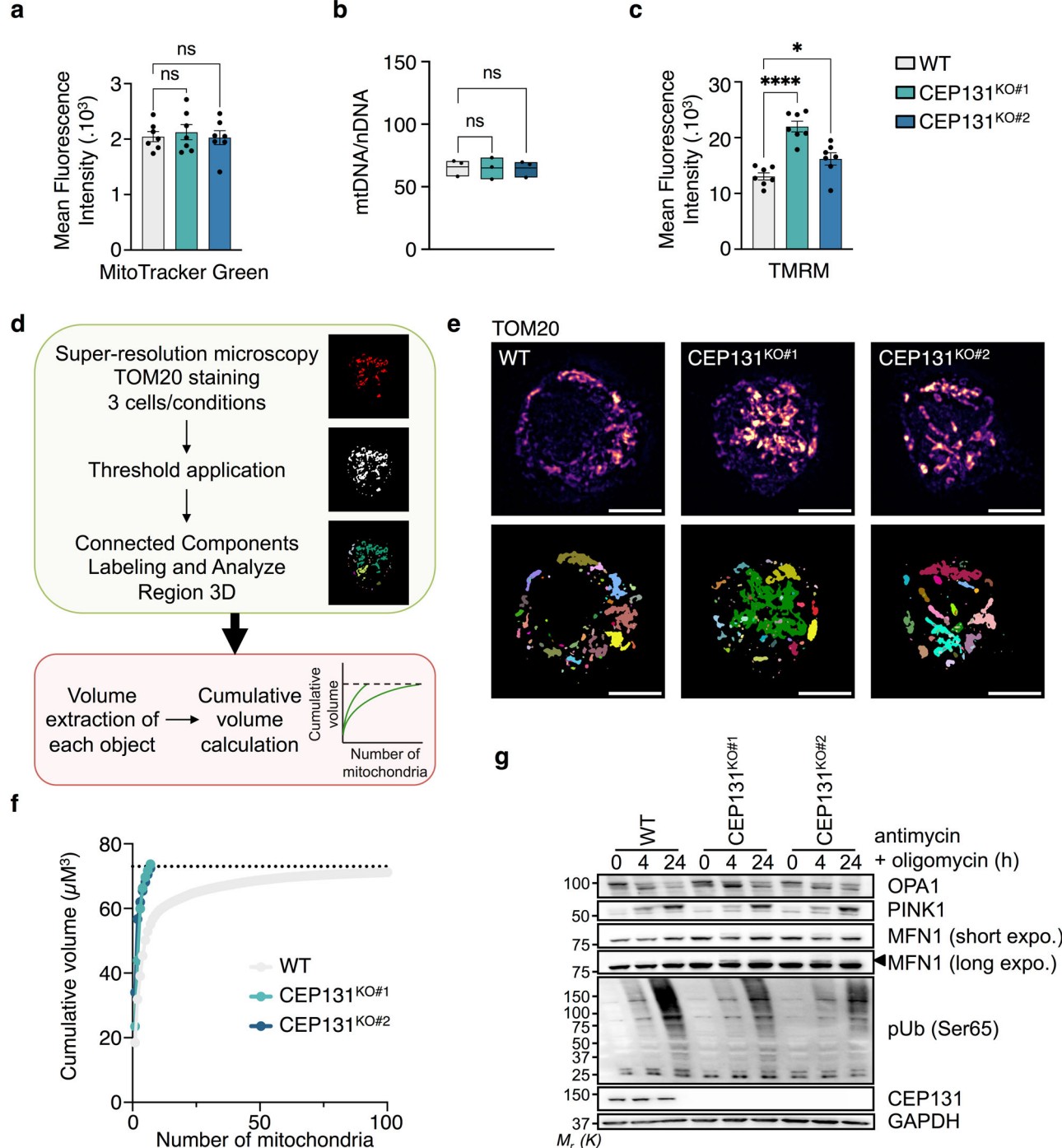

**Fig. 3 CEP131 participates in mitochondrial dynamics. a** Flow cytometry analysis of the mitochondrial mass using MitoTracker Green in wild-type (WT), CEP131 KO#1, and KO#2 Jurkat cells (mean ± SEM, $n = 7$ biological replicates, one-way ANOVA, ns: non-significant). **b** Expression of the mitochondrial DNA (mtDNA) normalized to the nuclear DNA (nDNA) assessed by qPCR. For mtDNA and nDNA, the expression of three genes were measured and the mean of three ratios mtDNA/nDNA is presented for $n = 3$ biological replicates (one-way ANOVA, ns: non-significant). **c** Flow cytometry analysis of the mitochondrial transmembrane potential using TMRM (tetramethylrhodamine) in WT, CEP131 KO#1, and KO#2 Jurkat cells (mean ± SEM, $n = 7$ biological replicates, one-way ANOVA, *$p < 0.05$, ****$p < 0.0001$). **d** Schematic overview of the workflow used to measure mitochondrial volumes. **e** TOM20 staining analysis by structure illumination microscopy (SIM) in WT, CEP131 KO#1, and #2 cells. Top panel, max intensity projection of z-stacked images are shown. Bottom panel, each detected object is represented by one color as explained in (**d**). Representative images of three independent experiments are shown (scale bars, 10 μm). **f** Cumulative plots of mitochondrial volume distribution in WT and CEP131 knockout cells derived from (**e**) (three cells/conditions). Data are representative of three independent experiments. **g** Cell lysates from WT, CEP131 KO#1, and KO#2 cells stimulated with 1 μM antimycin and 1 μM oligomycin for the indicated times were prepared and analyzed by immunoblotting with specific antibodies. Black arrowhead, ubiquitinated MFN1. GAPDH served as a loading control. Molecular weight markers ($M_r$) are shown. Data are representative of three independent experiments.

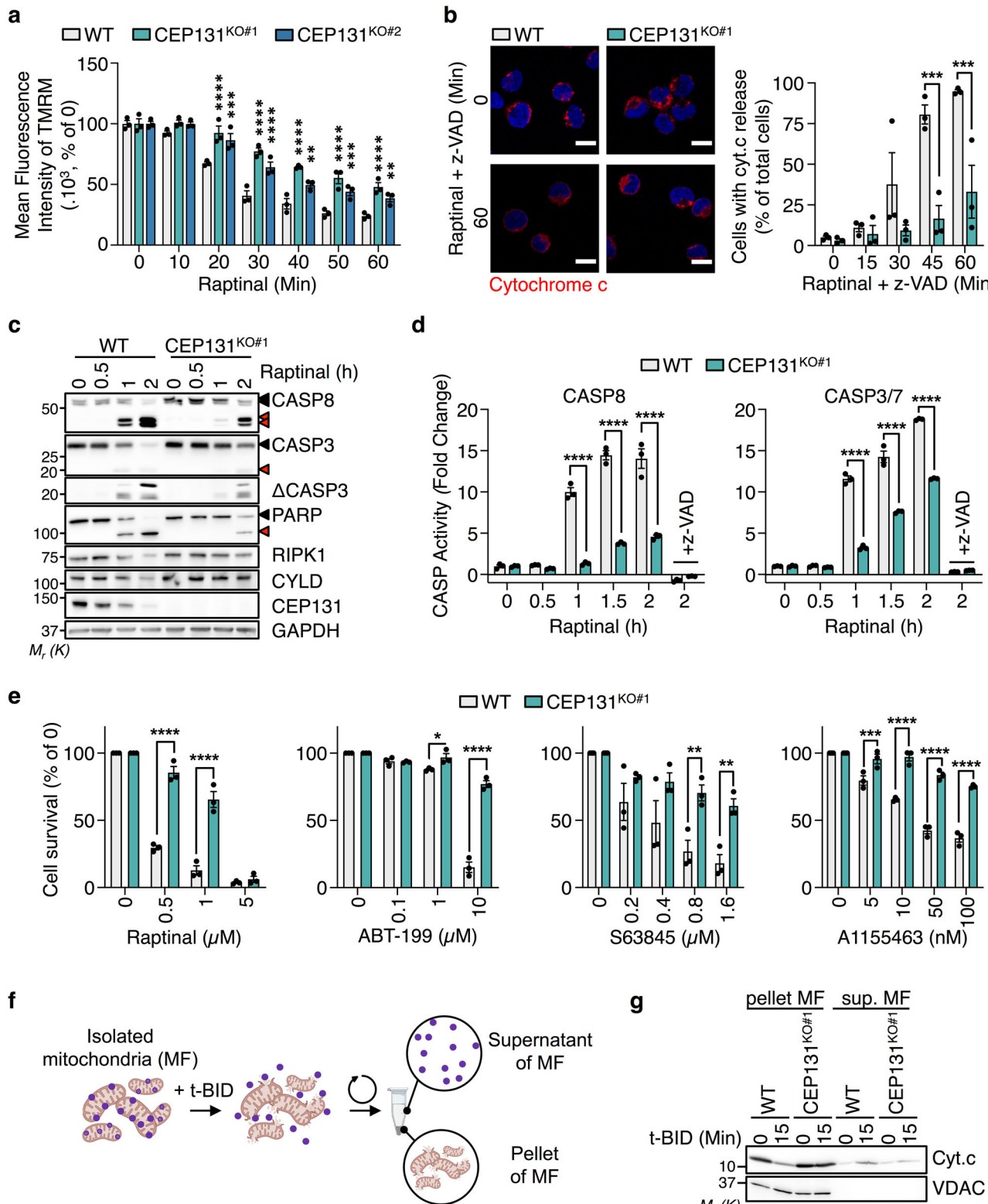

(Supplementary Fig. 3h). In addition, Raptinal-mediated cell death was also delayed in L929 when CEP131 is silenced (Supplementary Fig. 3j). Lastly, we wondered if this function we ascribed to CEP131 during apoptosis requires centriolar satellites. To this end, PCM1 was silenced in HT-29 cells. Paralleling our results with both CEP131 knockout clones, the silencing of PCM1 also delayed CASP activation and apoptosis (Supplementary Fig. 4a–c). Finally, we found that silencing PCM1 did not further

reduce CASP3 cleavage in CEP131 knockout Jurkat cells (Supplementary Fig. 4d). Collectively, these results supported a role for CEP131 and centriolar satellites in the timely permeabilization of the mitochondria and subsequent apoptotic cell death.

We noticed that during the execution phase of apoptosis, when CASPs are fully active, CEP131 and most of the centriolar satellite components tested were processed. These cleavages were

**Fig. 4 CEP131 participates in mitochondria-dependent cell death. a** Wild-type (WT), CEP131 KO#1, and KO#2 Jurkat cells were treated with 10 μM Raptinal for the indicated times and mitochondria membrane potential was assessed by flow cytometry analysis using TMRM (mean ± SEM, $n = 3$ biological replicates, two-way ANOVA, **$p < 0.01$, ***$p < 0.001$, ****$p < 0.0001$). **b** Left, confocal microscopy analysis of cytochrome c distribution (red) in cells treated as indicated. The pan-caspase inhibitor z-VAD was added 30 min prior to treatment with Raptinal. Nuclei were counterstained with 4′–6-diamidino-2- phenylindole (DAPI). Scale bars, 10 μm. Representative images of three independent experiments are shown. Right, quantification of the percentage of cells presenting cytochrome c release (mean ± SEM, $n = 3$ biological replicates, two-way ANOVA, ***$p < 0.001$). **c** Cell lysates from WT and CEP131 KO#1 cells stimulated with 10 μM Raptinal for the indicated times were prepared and analyzed by immunoblotting with specific antibodies. Full-length and cleaved protein forms are indicated with black and red arrowheads, respectively. GAPDH served as a loading control. **d** CASP8 and CASP3/CASP7 catalytic activity was measured by luminescent substrate cleavage (Caspase-Glo assay) in cells treated with 10 μM Raptinal for the indicated times. The pan-caspase inhibitor z-VAD (20 μM) was added 30 min prior to treating with Raptinal and served as a control (mean ± SEM, $n = 3$ technical replicates, one representative experiment of three independent one is shown, two-way ANOVA, ****$p < 0.0001$). **e** Cell viability was assessed by CellTiter Glo in cells treated with increasing concentrations of Raptinal, ABT-199, S63845, or A1155463 for 16 h in WT and CEP131 KO#1 cells (mean ± SEM, $n = 3$ biological replicates, two-way ANOVA, *$p < 0.05$, **$p < 0.01$, ***$p < 0.001$, ****$p < 0.0001$). **f, g** Fractions enriched with mitochondria (mitochondrial fraction, MF) from WT and CEP131 KO#1 cells were treated with 10 nM truncated BID (t-BID) at 30 °C for 0 and 15 min. After a centrifugation step, the supernatants and mitochondria pellets were collected and analyzed by immunoblotting with antibodies against cytochrome c (cyt.c). VDAC served as a control of mitochondria isolation. Data information: (**c**, **g**). Molecular weight markers are shown. Data are representative of three independent experiments.

efficiently prevented by treating cells with the pan-CASP inhibitor z-VAD, suggesting proteolysis by these proteases (Supplementary Fig. 5a, b). Interestingly, active cleaved species of CASP3 and CASP8 accumulated in a fraction enriched with the centrosome and centriolar satellites (Supplementary Fig. 5c). Our confocal microscopy analysis further showed a remodeling of CEP131 characterized by an aggregation around the centrosome in response to multiple cell death inducers (Supplementary Fig. 5d). In the case of TNFα-mediated cell death, this remodeling was blocked by the RIPK1 inhibitor Necrostatin-1, but not by z-VAD (Supplementary Fig. 5d). To test whether CEP131 might be a direct target of CASPs, we devised an in vitro cleavage assay by mixing immunopurified CEP131 or PCM1 proteins with recombinant human CASP3 and CASP8. This revealed that CASP3, and not CASP8, was able to cleave directly CEP131 and PCM1 (Supplementary Fig. 5e). Using the online PROSPERous tool (prosperous.erc.monash.edu), we identified putative CASP3 cleavage sites on CEP131 and performed site-directed mutagenesis. This showed that CEP131 was at least in part cleaved after the aspartic acid in position 548 (Supplementary Fig. 5f–h). Together, these data suggest that CEP131 and centriolar satellite components are remodeled during apoptosis, and are cleaved by CASP3.

Finally, we wondered whether the mitochondrial defect we observed in CEP131-depleted cells is intrinsic. To this end, we isolated mitochondria and tested, in vitro, their ability to release cytochrome c when exposed to recombinant truncated BID (t-BID), a BCL2-family member that permeabilizes the mitochondrial outer membrane when cleaved by CASP8[43] (Fig. 4f). As expected[38], t-BID drove a robust release of cytochrome c from the mitochondria pellet to the supernatant (Fig. 4g). By contrast, cytochrome c remained essentially in the mitochondria-enriched fraction of CEP131 knockout cells. These data suggest that the delayed release of cytochrome c from mitochondria and subsequent cell death observed in CEP131 knockout cells results from intrinsic changes in mitochondria.

The centrosomal and centriolar satellite element CEP131 has been linked to key cellular functions including primary cilia formation and aggresome clearance[18,25,44]. Yet, the landscape of its functions continues to be elucidated. By combining quantitative proteomics and functional analyses in a T-cell line, we now provide evidence that CEP131 also contributes to the maintenance of proteostasis, and finely tunes TCR signaling and mitochondrial apoptosis. In that view, defining whether CEP131 modulates the equipoise between life and death decisions in primary lymphocytes as well as in cells from different lineages will be of clear interest. How CEP131 controls these apparently

independent cellular functions remains to be unfolded. Hints may come from our observation that CEP131 participates in the optimal dynamics of microtubule and tubulin PTMs. The crucial role of the tubulin code on intracellular organelle distribution and positioning[45] may explain the elongated mitochondrial network and the change in endosomal trafficking we observed without CEP131. For instance, the contacts between the endoplasmic reticulum and mitochondria are known to influence the morphology of mitochondria and the transfer of lipids[46], thereby facilitating the interaction and function of the BCL2 family members and the regulation of mitochondrial permeabilization[47]. In the same vein, mitochondrial cardiolipin regulates the oligomerization and activity of DRP1, a large GTPase involved in mitochondria fission[48,49]. Therefore, it will be interesting to investigate the potential interplay between CEP131 and DRP1, as well as the impact of CEP131 on the lipid composition of mitochondria. The defect we observed in the recycling and trafficking of CD3ε without CEP131 draws an interesting parallel with the intraflagellar transport protein (IFT-20), another regulator of ciliogenesis, which forms a complex with Rab5a on early endosomes[31,50]. Whether CEP131 intersects with IFT-20 therefore warrants further investigation. Moreover, since the centrosome docks at the plasma membrane of lymphocytes during the formation of the immune synapse[51,52], it is tempting to speculate that centriolar satellites and CEP131 may participate more directly in TCR signaling. On a different note, mitochondrial dynamics and mitosis progression are intricately related[53], and the deletion of CEP131 was shown to slow cell proliferation, reduce the proportion of mitotic cells, and increase the rate of mitotic errors[21,23]. Although no overt change in the cell cycle phase was observed, Jurkat cells knockout for CEP131 displayed reduced cell proliferation, in line with previous works[21,23]. In that sense, defining whether CEP131 impacts mitosis and if this contributes to mitochondrial distribution would be of interest.

CEP131 partition between the centrosome and centriolar satellites and whether pools of CEP131 with specific functions exist remains unclear. In keeping with this idea, only a subset of PCM1 is involved in the GABARAP-dependent autophagy[17]. An additional layer of intricacy emerges from the dynamic and versatile positioning of CEP131 depending on the cellular context[54]. For instance, CEP131 accumulates at the centrosome of mitotic cells and cells lacking centriolar satellites[9,21]. In that view, our findings that PCM1 silencing phenocopies CEP131 deletion may be attributed to a mislocation of CEP131 and/or reduced abundance of CEP131[9,22]. Interestingly, cellular stresses such as UV radiations trigger MK2-mediated phosphorylation of CEP131, subsequently causing its sequestration in the cytosol[5,6].

We observed that CEP131 remains essentially peri-centrosomal during apoptosis, although it appears more clustered. Whether this reflects aggregation or whether a pool of peripheral particles relocates will require additional studies. Finally, we report that, alongside ubiquitination and phosphorylation[54], CEP131 is subject to another level of post-translational modification with proteolysis when cells are fully engaged in apoptosis. We further conclusively identified CASP3 as the protease that targets CEP131. Interestingly, several centriolar satellite components were also cleaved during apoptosis. Although we cannot exclude that this cleavage inactivates CEP131 to limit cell death, our results echo the recent report by Seo and Rhee that the centrosomal proteins SAS-6 and pericentrin are trimmed by CASPs during apoptosis[55]. Such cleavages may favor the apoptotic process through microtubule destabilization and the formation of an apoptotic microtubule network[55]. Future work will be aimed at defining the functional consequences of CEP131 and centriolar satellite components cleavage during apoptosis.

## Methods

**Cell culture and reagents**. Jurkat E6.1 T lymphocytes, L929 cells, HeLa cells, and HT-29 cells were purchased from American Type Culture Collection (ATCC). Cells were grown with RPMI1640 (Jurkat E6.1, Eurobio Scientific), DMEM (HeLa, L929, Life Technologies), or with Mc Coy's 5a (HT-29, Life Technologies) supplemented with 10% Fetal Bovine Serum, Glutamax (Life Technologies), Penicillin/Streptomycin (Life Technologies) and normocin (Invivogen). HEPES (Life Technologies) and Sodium Pyruvate (Life Technologies) were added to Jurkat E6.1 media. All cell lines were maintained at 37°C with 5% $CO_2$. Cells were frequently tested for mycoplasma contamination. For the proliferation assay, cells were plated in a 12-well plate at 0.1 or $0.5.10^6$ cells.mL$^{-1}$ and were counted each day for 5 days. The cell cycle analysis was assessed using the NucleoCounter® NC-3000 fluorescent imaging cytometer and the two-step cell cycle analysis.

Antimycin A (A8674), Cycloheximide, Etoposide, Ionomycin, Oligomycin (O4876), Phorbol 12,13-dibutyrate (PDBu), Phorbol 12-Myristate 13 Acetate (PMA), Protein A, Raptinal and Staurosporine were purchased from Sigma. A-11 (A1155463) (S7800), ABT-199 (S8048), Birinapant (S7015), Necrostatin-1s (S8037), Nocodazole (S2775), Reversin (S7588), S63 (S63845) (S8383), z-VAD-fmk (S8102) were obtained from Selleckchem. Anti-CD3 (BD Pharmingen), anti-CD28 (BD Pharmingen), anti-Fas (Apo-1-3, COGER), anti-human CD3 UCHT1 (300438, Biolegend), Caspase Glo -3/7 (Promega), Caspase-8 Glo (Promega), CellTiter Glo (Promega), MG-132 (Cell Signaling Technology), MitoTracker Green (MTG) (M7514, Invitrogen), polyclonal goat anti-mouse (553998, BD Biosciences), recombinant human Caspase-3 (R&D), recombinant human Caspase-8 (R&D), TMRM (Invitrogen) and TNFα (R&D) were also used.

**siRNA and transfection**. HT-29 and L929 cells were transfected with 10 pmol of siRNA according to the manufacturer's instructions, using the Lipofectamine RNAiMAX Transfection Reagent (Invitrogen). E6.1 Jurkat cells were transfected with 10 pmol of siRNA by electroporation (BTX ECM830, Harvard Apparatus; 10 ms, 300 V). The following siRNA sequences were used: control low GC (12935111; Life Technologies), PCM1#2 (HSS107660; 5'-ACAGGUCCUACAACGUGACUUUAAA-3'), PCM1#3 (HSS107661; 5′-GCCUAACCCUUUGCCGUUACGUU UA-3′), CEP131#2 (5′-GUUGAGAUGCCCACGGCUAUU-3′), and mouse CEP131#1 (5′-AGACACAGGGCUAAGGGUA-3′)[25].

**CRISPR/Cas9 knockout and mutagenesis**. Single-guide RNA (sgRNA) targeting CEP131 was cloned into a lentiviral lenti-CRISPRv2 (GeCKO; ZhangLab) backbone[56,57]. The sgRNA sequence used is: 5′- CACCGAAGATCCAACAGCACCACGC-3′. Lentiviral particles were produced in HEK293T by co-transfection of the construct with p-VSV-G and pPAX2 plasmids using a standard calcium phosphate protocol. The lentiviral particles-containing supernatants were collected after 48 h and added on E6.1 Jurkat cells during a 1250 x $g$ centrifugation for 90 min in the presence of 8 µg.mL$^{-1}$ of polybrene (Santa Cruz). 1 µg.mL$^{-1}$ of puromycin was used to select infected cells. After limit dilution, viable single clones were amplified and tested by immunoblotting to control for CEP131 negative expression, and CEP131 knockout was confirmed by genomic DNA sequencing after PCR amplification of CRISPR mutation region. Briefly, gDNA was extracted with Nuclesospin Tissue DNA kit (Macherey-Nagel) following the manufacturer's recommendations, and amplified by PCR using Platinum PCR SuperMix High-fidelity kit (Invitrogen) with primers surrounding predicted mutation locus (forward: 5′-AC TGCCTTTGAGGCCACTT-3′, reverse: 5′-GGTCACTCCAGGCC TCACTA-3′). Sanger sequencing was performed on amplicons and two bi-allelic knockout clones were further selected. A list of in silico predicted off-targets is also shown using CRISPOR online software in Supplementary Table 1. Site-directed mutagenesis was performed on IRES-puro-3FLAG-CEP131 plasmid[24] to substitute the aspartic acid (D) residue on 496 or 548 with an alanine (A).

**Sample preparation, liquid chromatography-coupled mass spectrometry analysis (nLC-MS/MS), protein identifications and quantifications, and analysis**. Cell pellets were solubilized in lysis buffer (2% SDS, 200 mM Tris-HCl, pH 8.5) and boiled 5 min at 95 °C. Samples were also reduced and alkylated (10 mM TCEP, 50 mM CAA). Thirty milligrams of each protein extracts were digested using trypsin (Promega) and S-Trap Micro Spin Column was used according to the manufacturer's protocol (Protifi, Farmingdale, NY, USA). Peptides were then speed-vacuum dried. nLC-MS/MS analyses were performed on a Dionex U3000 RSLC nano-LC- system coupled to a TIMS-TOF Pro mass spectrometer (Bruker Daltonik GmbH, Bremen, Germany). After drying, peptides were solubilized in 30 µL of 0.1% TFA containing 10% acetonitrile (ACN). Two µL were loaded, concentrated and washed for 3 min on a C18 reverse phase column (5µm particle size, 100 Å pore size, 300 µm inner diameter, 0.5 cm length, from Thermo Fisher Scientific). Peptides were separated on an Aurora C18 reverse phase resin (1.6 µm particle size, 100 Å pore size, 75µm inner diameter, 25 cm length mounted to the Captive nanoSpray Ionisation module, from IonOpticks, Middle Camberwell Australia) with a 4 h run time with a gradient ranging from 98% of solvent A containing 0.1% formic acid in milliQ-grade H2O to 35% of solvent B containing 80% acetonitrile, 0.085% formic acid in mQH2O. The mass spectrometer acquired data throughout the elution process and operated in DIA PASEF mode with a 1.38 second/cycle, with Timed Ion Mobility Spectrometry (TIMS) enabled and a data-independent scheme with full MS scans in Parallel Accumulation and Serial Fragmentation (PASEF). Ion accumulation and ramp time in the dual TIMS analyzer were set to 100 ms each and the ion mobility range was set from 1/K0 = 0.63 Vs cm-2 to 1.43 Vs cm-2. Precursor ions for MS/MS analysis were isolated in positive polarity with PASEF in the 100-1.700 m/z range by synchronizing quadrupole switching events with the precursor elution profile from the TIMS device. The mass spectrometry data were analyzed using DIA-NN version 1.8.1[58]. The database used for in silico generation of spectral library was a concatenation of Human sequences

from the Uniprot and-Swissprot databases (release 2022-05) and a list of contaminant sequences. Oxydation of methionines was set as variable modification and carbamidomethylation of cysteins was set as permanent modification and one trypsin misscleavage was allowed. Precursor false discovery rate (FDR) was kept below 1%. The "match between runs" (MBR) option was allowed. Enrichment analyses of the proteomic experiment were performed using the STRING public online database (https://string-db.org). Proteomic data have been deposited to the ProteomeXchange Consortium via PRIDE and are available via ProteomeXchange with the identifier PXD042470.

**Depmap portal analysis**. The top-100 co-dependencies genes with PCM1 or CEP131 were extracted from the CRISPR dataset (DepMap Public 22Q4+Score, Chronos) downloaded from the Depmap portal (https://depmap.org/portal/). The identified genes were processed using the STRING public online database (https://string-db.org).

**Lymphocyte activation**. Cells were either stimulated with 20 ng.mL$^{-1}$ Phorbol 12-Myristate 13 Acetate (PMA) plus 300 ng.mL$^{-1}$ ionomycin, or with antibodies against CD3 and/or CD28 (1 μg.mL$^{-1}$ each). In some experiments, cells were placed on ice prior to incubation with 10 μg.mL$^{-1}$ of mouse anti-CD3 antibodies (UCHT1) for 20 min. After 2 washes with ice-cold PBS, CD3 was crosslinked by adding 5 μg.mL$^{-1}$ of polyclonal goat anti-mouse antibodies at 4 °C. Cells were then placed at 37 °C and collected at different times.

**Cell death induction, measurement of caspase activity, and cell viability**. The apoptosis inducer Raptinal was used either at 10 μM for short-time stimulation or at 0–5 μM for 16 h. For TNFα-induced cell death, cells were pre-treated for 30 min with 5 μM Birinapant prior to stimulation with 0-10 ng.mL$^{-1}$ of TNFα[59]. To stimulate the Fas receptor, anti-Fas antibodies (Apo-1-3; 0–1 ng.mL$^{-1}$) were combined with Protein A (0–1 ng.mL$^{-1}$). In some experiment, ABT-199 (0–1.10$^4$ nM), S63845 (0–1600 nM), A1155463 (0–100 nM), staurosporine (STS; 0–500 nM), or etoposide (0–40 μM) were used. Caspase-8 and Caspase-3/−7 activities were measured using Caspase-8 Glo and Caspase-3/−7 Glo (Promega) respectively, following the manufacturer's instructions. Cell viability was assessed using CellTiter-Glo (Promega) following the manufacturer's instructions.

**Lysates preparation, cell fractionation, immunoprecipitation, in vitro assays, and immunoblotting**. Cells were washed twice with ice-cold PBS prior to lysis with RIPA buffer [25 mM Tris-HCl (pH 7.4), 150 mM NaCl, 0.1% SDS, 0.5% Na-Deoxycholate, 1% NP-40, 1 mM EDTA] supplemented with 1X Halt Protease Inhibitor cocktail (ThermoFisher Scientific), for 30 min on ice. Samples were cleared by centrifugation at 10,000 x g prior to determination of protein concentration by a BCA kit (UP40840A; Interchim). 5–10 μg of proteins were incubated with 2X Laemmli buffer (Life Technologies) at 95°C for 3 min. Lysates were resolved by SDS–polyacrylamide gel electrophoresis (SDS–PAGE) using 3-15% Tris-Acetate or 5-20% Tris-Glycine gels and transferred onto nitrocellulose membranes (GE Healthcare).

For immunoprecipitation, cells were lysed with TNT buffer [50 mM Tris-HCl (pH 7.4), 150 mM NaCl, 1% Triton X-100, 1% Igepal, 2 mM EDTA] supplemented with 1X Halt Protease Inhibitor cocktail and samples were precleared with Protein G Sepharose (Sigma) for 30 min, prior incubation with 1 μg of antibodies and Protein G Sepharose for 2 h or 16 h at 4 °C with rotation. Nonrelevant Antibodies served as a control for PCM1. For recombinant caspases assay, immunoprecipitates were incubated with recombinant human caspase-3 or −8 in assay buffer (25 mM HEPES, 0,1 % (w/v) CHAPS, 10 mM DTT) for 1 h at 37°C.

For cell fractionation, 20.10$^6$ cells were washed with ice-cold PBS prior to resuspension in fractionation buffer [20 mM Hepes (pH 7.5), 1.5 mM MgCl$_2$, 60 mM KCl] supplemented with 1X Halt Protease Inhibitor cocktail for 5 min. After 20 passages in a 29 G syringe, samples were centrifuged at 1000 × g for 5 min. Supernatants were then centrifuged at 5000 × g, 10,000 × g, and 20,000 × g for 10 min to obtain the P5, P10, and P20 pellets and the S20 fraction. After one wash with fractionation buffer, pellets were lysed with RIPA buffer supplemented with 1X Halt Protease Inhibitor cocktail for 30 min.

Antibodies specific for the following proteins were purchased from Santa Cruz Biotechnology: CD3ε (SC-1179), CD3ζ (SC-1239), CEP290 (SC-390462), CYLD (SC-137139), DRP1 (SC-271583), GAPDH (SC-47724), MIB1 (SC-393811), PARK8 (sc32282), PARP (SC-8007), PCM1 (SC-398365), TBK1 (SC-73115), TCRα (SC-51571), γ-Tubulin (SC-51715). Antibodies against ATM (2873), caspase-3 (9662), cleaved caspase-3 (9664), caspase-8 (9746 L), CHK2 (3440), DYKDDDDK Tag (FLAG-M2; 8146), MFN1 (14739), pATM (5883), pCHK2 (2661), pERK (9106 S), pIκBα (S32/36) (9246), PINK1 (6946), PLCγ1 (5690), pPLCγ1 (14008), pp65 (S536) (11/17), and RIPK1 (3493) were purchased from Cell Signaling Technology. Antibodies specific for the following proteins were obtained from Proteintech: BBS4 (12766-1-AP), CEP131 (C-ter, 25735-1-AP), cytochrome c (10993-1-AP), GABARAPL1 (11010-1-AP), OPA1 (66583-1-AP), α-Tubulin (66031-1-1 g), β-Tubulin (66240-1-1 g). Antibodies against CCDC66 (A303-339A; Bethyl), CEP72 (A301-297A, Bethyl), CEP131 (N-ter, A301-415A, Bethyl), CEP290 (A301-659A, Bethyl), ERK (ab218017, Abcam), glycylated tubulin (AG-25B-0034-C100, Coger), GSDME (ab215191, Abcam), polyglutamate chain (polyE, AG-25B-0030-C050, Coger), polyglutamylation modification tubulin (AG-20B-0020-C100, Coger), PCM1 (ab72443, Abcam), OFD1 (HPA031103, Atlas antibodies), and phospho-tyrosine (OR-321X, Millipore) and phospho-ubiquitin Ser65 (ABS1513-I) were also used.

**Mitochondria isolation and in vitro permeabilization assays**. Intact mitochondria-enriched fractions were purified as previously described[38]. In brief, cells were washed twice in ice-cold PBS and centrifuged at 1000 × g for 2 min. The cell pellet was resuspended in 5 mL of HIM Buffer (200 mM mannitol, 70 mM sucrose, 1 mM EGTA, 10 mM HEPES) supplemented with 1X Halt Protease Inhibitor cocktail and centrifuged at 1000 × g for 5 min. Pellets were resuspended in 1 mL of HIM Buffer supplemented with 1X Halt Protease Inhibitor cocktail and passed through a 29 G syringe 10 times. Samples were centrifuged for 10 min at 600xg and the supernatants were subsequently centrifuged for 15 min at 7000 × g to obtain the post-mitochondria supernatant (PNS) and the mitochondria fraction (MF, pellet). The MF was resuspended in 300 μL HIM Buffer supplemented with 1X Halt Protease Inhibitor cocktail. 30 μg of MF were incubated with 10 nM of truncated-BID (t-BID, R&D Systems) in KCl buffer (125 mM KCl, 4 mM MgCl$_2$, 5 mM Na$_2$HPO$_4$, 5 mM succinate, 0.5 mM EGTA, 15 mM Hepes, 5 μM rotenone) for different times at 30 °C. Mitochondria were recovered by centrifugation at 13,000 × g for 5 min. The supernatants were kept (sup. MF) and the pellets (pellet MF) were lysed with 20 μL RIPA buffer supplemented with 1X Halt Protease Inhibitor cocktail for 30 min. Samples were cleared by centrifugation at 10,000 x g prior determination of protein concentration. 5 μg of pellet MF and the same volume for the corresponding supernatant MF were collected for immunoblotting analyses.

**Immunofluorescence**. Cells were fixed in 4% paraformaldehyde-PBS for 12 min at room temperature. Samples were permeabilized and blocked using 4% BSA- 0.3% Triton X-100- PBS for 1 h at room temperature. Primary and secondary antibodies were incubated in the blocking solution for 1 h at room temperature. For centriolar satellite staining, cells were incubated with ice-cold methanol for 2 min and then quenched for 10 min with 100 mM glycine-PBS. Primary and secondary antibodies were incubated in 0.2% BSA-0.05% Saponin-PBS for 1 h at room temperature[22]. Microtubule regrowth assay was performed as previously described[19]. Briefly, cells were treated with 10 µM nocodazole for 1 h at 37°C, washed once with fresh media, and placed in fresh media to allow microtubule regrowth. After 5 min, cells were fixed with 4% paraformaldehyde in PEM buffer (80 mM PIPES pH 6.9, 2 mM $MgCl_2$, 5 mM EDTA, 0.5% Triton X-100) for 12 min at room temperature, permeabilized with 0.3% Triton X-100-PBS and blocked in 3%BSA-PBS. Primary antibodies were incubated with blocking solution overnight at 4°C and with secondary antibodies for 1 h at room temperature. Nuclei were counterstained with 4′,6-diamidino-2-phenylindole (DAPI), and samples were mounted using Prolong gold anti-fade mounting media (Life Technologies). Confocal images were acquired with a Nikon A1R confocal microscope using the NIS-Element software. For structure illumination microscopy (SIM), images were acquired on a Nikon N-SIM microscope (MicroPicell Facility, SFR François Bonamy, France). The reconstruction of images in 3D was done using the NIS-Element Software from Z stacks of 0.12 µm taken using a 100x oil-immersion lens with a 1.49 numerical aperture. Images were processed using FIJI software. Mitochondria volumes were determined using MorphoLibJ plugin[60]. Briefly, a threshold using Yen method was applied on Z stack SIM images. 3D images were then processed using "Connected Components Labeling" module with "connectivity=26 type = [16 bits]" options. "Analyse Regions 3D" (method = [Crofton (13 dirs.)], euler_connectivity=26") module was used to characterize detected objects per individual cells and volumes of each object were extracted. For the cumulative volume plot, volumes of individual mitochondria were cumulated in descending order for 3 cells per condition. The following antibodies were used: CEP72 (A301-297A, Bethyl), CEP131 (A301-415A, Bethyl), CEP290 (IHC-00365-2, Bethyl), cytochrome-c (SC-13561, Santa Cruz), MIB1 (M5948, Sigma-Aldrich), PCM1 (5213 S, Cell Signaling), α-Tubulin (66031-1-1 g, Proteintech), γ-Tubulin (T6557, Sigma-Aldrich).

**Flow cytometry**. Mitochondrial mass was measured after incubating cells with 12.5 nM of MitoTracker Green (MTG) for 1 h at 37 °C. For mitochondrial membrane potential, Tetramethylrhodamine, methyl ester (TMRM) was used at 0.1 µM for 30 min at 37°C before flow cytometry analysis. The staining of surface proteins was performed on living cells with antibodies incubated for 1 h at 4°C. Intracellular staining was performed using BD Cytofix/Cytoperm Kit (BD Biosciences) following the manufacturer's protocol. For the CD3ε recycling experiment, cells were treated with 1 µM Phorbol 12,13-dibutyrate (PDBu) for 1 h at 37 °C, washed at 4 °C and put at 37°C for 0–60 min, as previously described[31]. CD3 expression at the surface was measured by flow cytometry at 5, 10, 15, 30, and 60 min, and normalized by total surface CD3ε. Flow cytometry analyses were performed on FACS Calibur (BD Biosciences; Cytocell Facility, SFR François Bonamy, France) and processed using FlowJo V10 software. Specific antibodies were used: FITC anti-human TCRα/β (BD Pharmingen, 347773), AlexaFluor 647 anti-human CD3ζ (CD247, BD Pharmingen, 566651), FITC anti-human CD3ε (BD Pharmingen, 555332), APC anti-human CD28 (BD Pharmingen, 559770). Details of the gating strategy are shown in Supplementary Fig. 6.

**qPCR and qPCR Array**. RNA was extracted using Nucleospin RNAplus kit (Macherey-Nagel), on three biological replicates ($4.10^6$ Jurkat cells). Equal amounts of RNA were reverse transcribed using the Maxima Reverse transcription kit (Thermo Scientific), and 25 ng of the resulting cDNA was amplified by qPCR using PerfeCTa SYBR Green FastMix (QuantaBio). Data were analyzed using the 2-ΔΔCt method and normalized by the housekeeping genes ACTB and HPRT1. The following primers were used: ACTB forward, 5′-GGACTTCGAGCAAGAGATGG-3′; ACTB reverse, 5′-AGCACTGTGTTGGCGTACAG-3′; HPRT1 forward, 5′-TGACACTGGCAAAACAATGCA-3′; HPRT1 reverse, 5′-GGTCCTTTTCACCAGCAAGCT-3′; CD3ε forward, 5′- TGCTGCTGGTTTACTACTGGA-3′; CD3ε reverse, 5′- GGATGGGCTCATAGTCTGGG-3′; CD247 (CD3ζ) forward, 5′- GATCCTGGGAGAAGGGACTC-3′; CD247 (CD3ζ) reverse, 5′-ATTCCATCCAGCAGGTAGCAG-3′; CD28 forward, 5′- CTATTTCCCGGACCTTCTAAGCC-3′; CD28 reverse: 5′- GCGGGGAGTCATGTTCATGTA-3′; CEP131 forward: 5′- CTTCATTGTCACCGGGCCTC-3′; CEP131 reverse: 5′- CCTTTCATGGTGGACAAGGC-3′. For the nDNA/mDNA analysis, total DNA was extracted using Nucleospin DNA Tissue kit (Macherey-Nagel), on three biological replicates ($4.10^6$ Jurkat cells). qPCR was performed as described below with 3 mitochondrial targets (CO2 forward, 5′- AGAGCACAGATACCCAGAACT-3′, reverse, 5′- GGTGATTCAGTGTGTCTTCCATT-3′, CYB forward, 5′- CAACATCTCCGCATGATGAAA-3′, reverse, 5′- CCATAATTTACGTCTCGAGTGATGTG-3′, ND1 forward, 5′-ATGGCCAACCTCCTACTCCT-3′, reverse, 5′- GCGGTGATGTAGAGGGTGAT-3′) normalized by 3 nuclear targets (18 S forward, 5′- CATTCGAACGTCTGCCCTATC-3′, reverse, 5′- CCTGCTGCCTTCCTTGGA-3′, ACTB forward, 5′- ACCCACACTGTGCCCATCTAC-3′, reverse, 5′- TCGGTGAGGATCTTCATGAGGTA-3′, GAPDH forward, 5′- GGAACCTCTCCTGGTCCTGTTG-3′, reverse, 5′- GTCCCCGCACCTCCAGAAAC-3′). The ratio between mtDNA and nDNA using 2ΔCt method was used to appreciate mitochondrial number of copies per analyzed conditions. The RT[2] profiler PCR array of human NF-κB signaling targets was performed on cells treated with 1 µg.mL[−1] of anti-CD3 and anti-CD28 for 2 h, as previously described[61]. RNA was extracted using the Nucleospin RNAplus kit (Macherey-Nagel), following the manufacturer's instructions. 2 µg of RNA was reverse transcribed using the Maxima Reverse Transcriptase kit (ThermoFisher). The RT[2] profiler PCR array of human NF-κB signaling targets was executed following the manufacturer's instructions (PAHS-225Z, Qiagen). The list of target genes can be found in Supplementary Table 2.

**Mitochondrial activity**. A Seahorse XFp Bioanalyser (Agilent) was used to determine OCR (Oxygen Consumption Rate) and ECAR (ExtraCellular Acidification Rate). Cells were washed in assay media (DMEM Base media (Sigma) with glucose (11 mM), sodium pyruvate (1 mM), and L-glutamine (2 mM), pH 7.4) before being plated onto Seahorse cell culture plates coated with Cell-Tak (Corning) at $3.10^5$ cells per well. After adherence and equilibration, OCR was measured during a Mito Stress assay with the addition of oligomycin (ATPase inhibitor, 1 µM), carbonyl cyanide 4-(trifluoromethoxy) phenylhydrazone (synthetic uncoupler CCCP; 1.2 µM) and antimycin A and rotenone (inhibitors of complex 3 and 1 respectively, 1 µM each). Assay parameters were as follows: 4 cycles of 3 min mix followed by 3 min measurement, at basal and after each injection. All values were normalized per 100,000 cells. Non-mitochondrial respiration was subtracted from basal and maximal OCR values. Calculation of coupling efficiency and Spare capacity (as %) was calculated as (ATP production rate/Basal respiration rate)*100, and (max/OCR)*100 as described by the manufacturer (Agilent).

**Statistical analysis and reproducibility.** Densitometry quantifications were performed using the ImageJ software. Statistical analyses were performed using GraphPad Prism 8 software using one-way and two-way analysis (ANOVA). The significance and number of repeats are indicated in figure legends.

**Reporting summary.** Further information on research design is available in the Nature Portfolio Reporting Summary linked to this article.

## Data availability

All data supporting the findings of this study are included in the article and its Supplementary information. Numerical source data for all charts and graphs can be found in Supplementary Data 1. The uncropped blots can be found in Supplementary Fig. 7. The mass spectrometry data from this publication have been deposited to the PRIDE database and assigned the identifier [PXD042470]. All other data are available from the corresponding author upon reasonable request.

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

## Acknowledgements

We thank Johanna Bruce, Morgane Le Gall, Marjorie Leduc, and Emilie-Fleur Gautier (Proteom'IC facility, University Paris Cité, CNRS, INSERM, Institut Cochin, F-75014 PARIS, France) for the expertise and technical support for proteomic analysis; Perrine Paul-Gilloteaux and the MicroPicell facility (SFR Santé François Bonamy, Nantes, France) for the expertise and discussions for imaging analysis; Yongfeng Shang and Lei Shi (Tianjin Medical University, China) for providing reagents; Sophie Barillé-Nion (CRCI²NA, Nantes, France) and Benjamin Vitré (CRBM, Montpellier, France) for helpful discussions. This work was supported by the DIM Thérapie Génique Paris Ile-de-France Région, IBiSA, and the Labex GR-Ex. This work was funded by an International Program for Scientific Cooperation (PICS, CNRS), Fondation de France, Fondation pour la Recherche Médicale (Equipe labellisée DEQ20180339184), Fondation ARC contre le Cancer (PJA to JG and NB), Institut National du Cancer (INCa_18384, INCa PLBIO 2019-151, INCa PLBIO 2019-291, INCa PAIR-CEREB lNCa_16285), Ligue nationale contre le Cancer (Equipe labellisée) et comités de Loire-Atlantique, Maine et Loire, Vendée, Ille-et-Vilaine (JG, NB). CCNR received a fellowship from INSERM, Région Pays de la Loire, and the Ligue Nationale contre le Cancer. JJ is supported by the Fondation de France and LM by the Ligue Nationale contre le Cancer. The team is part of the SIRIC ILIAD (INCA-DGOS-INSERM-ITMO Cancer_18011).

## Author contributions

Conceptualization: C.C.N.R., K.T., J.J., L.M., O.R., M.L.B., C.P., J.G., N.B.; Methodology: C.C.N.R., K.T., J.J., L.M., O.R., M.L.B.; Investigation C.C.N.R., K.T., J.J., L.M., O.R., M.L.B., Z.C.; Visualization: C.C.N.R., K.T., J.J., L.M., O.R., M.L.B., C.P., J.G., N.B.; Funding acquisition: J.G., N.B.; Supervision: C.P., J.G., N.B.; Writing—original draft: C.C.N.R., N.B.; Writing—review & editing: C.C.N.R., K.T., J.J., O.R., C.P., J.G., N.B.

## Competing interests

The authors declare no competing interests.
