## [Peer Review File · Communications Biology]

Reviewers' comments:

Reviewer #1 (Remarks to the Author):

In this manuscript, Renaud et al. elucidated the non-ciliary functions of Cep131 during signaling and cell death. To this end, they generated Cep131 knockout Jurkat T cell lines and characterized two Cep131 KO clones using a set of proteomic and functional assays. Cep131 KO lines were defective in proteostatic regulation, autophagy, optimal antigen receptor signaling and mitochondrial apoptosis. Another siRNA-mediated depletion of another centriolar satellite component PCM1 also resulted in also delayed mitochondrial cell death associated with cytochrome c release and CASP activation, suggesting a potential new function for centriolar satellites in these processes.

Cep131 was previously characterized as a centrosome and centriolar satellite proteins during a wide range of biological processes using cellular and organismal models. This manuscript advances our knowledge on CEP131 mechanisms and functions by uncovering its functions in mitochondrial apoptosis. Although other functions such as autophagy and tubulin PTMs have been assigned to CEP131 in this study, mitochondrial apoptosis is well-characterized for the specific defects caused by CEP131. Since the authors used a T cell line that does not form a primary cilium, they could specify the CEP131 functions independent of primary cilium, which is a strength of this manuscript. Overall, the presented data in the manuscript is of high quality and support most of the major conclusions of the paper. The manuscript will benefit from revision of the introduction to introduce more details on CEP131 functions derived from different models as well as to the biology of centriolar satellites. To sum up, the manuscript will be of general interest to the cell biologists, particularly for those studying centrosomes/centriolar satellites, cell death and signaling and I recommend publication after my comments below are addressed:

1- CEP131 localizes both to the centrosome and centriolar satellites. In some parts of the manuscript, the authors concluded or discussed the phenotypes of CEP131 as centriolar satellite phenotypes, which should be revised as the phenotypes might also be linked to centrosomal pools of CEP131. It will be misleading to assign the new CEP131 phenotypes as centriolar satellite-specific phenotypes except for CASP activation and delayed apoptosis resulting from depletion of centriolar satellite marker PCM1 in Fig. S3.

2- To determine the functional relationship between centriolar satellites and CEP131, CASP activation and apoptosis assays performed in CEP131 Kos or PCM1 depleted cells should be performed in cells depleted/deleted for both proteins. Comparative analysis of reported phenotypes with that of CEP131 KO depleted for PCM1 will allow authors to assess how loss of centriolar satellites (PCM1 depletion) changes CEP131-linked phenotypes.

3- Along the lines of CEP131 having centrosome and centriolar satellite pools, the title should be revised to better represent the paper. Cep131 localizes to both centrosome and centriolar satellites, so this should be included. "timely" can be removed from the title.

4- Can authors compare the results from western blot analysis of Cep131 KO lines in Fig. 1 with that of global proteomics in Fig. 2? This will be a good way for validating the proteomics data. Moreover, densitometry analysis for Fig. 1A is required to quantitatively report the changes in the cellular abundance of the centriolar satellite proteins in CEP131 KO cells. Although CCDC66 levels were concluded to not change, the blots do not represent this conclusion.

5- Characterization of CEP131 KO lines for potential changes in PTMs in Fig. 1F was only performed by western blot analysis with antibodies against different PTMs, which is not sufficient to conclude that CEP131 functions in this process. The emphasis on this data throughout the manuscript should be tuned down. This data can also be moved to the supplemental figures. Similar comments also apply to autophagy function in Fig. 1j.

6- Densitometry analysis is required for Fig. 4a as some conclusions for this figure do not match with the representative data.

7- Supp. Fig. 4d includes only one cell micrography per condition to represent changes to CEP131

cellular distribution. This phenotype should be quantified like it was done in Fig. 1b,c.

8- The authors did not perform phenotypic rescue experiments with CEP131 to confirm the specificity of phenotypes. I understand that the characterization of two different CEP131 clones is chosen instead to show specificity. However, these clones were only validated by western blotting in Fig. 1. To show that these clones do not have the same CEP131 mutation after genome editing, sequencing of the clones is required, which will be important to show specificity.

Minor comments:

1- Pg.1 Introduction: Satellites move in a motor-dependent manner (both kinesin and dynein). Please revise the following sentence:

Centriolar satellites are small non-membranous proteinaceous particles of 70-100 nm that crawl along microtubules in a dynein-dependent manner and gravitate toward the centrosome

2- Fig. 1g: One datapoint for gly. tubulin is very different than the rest two. To be conclusive, it might be worth increasing the number of experimental replicates.

3- Abstract: "optimal tubulin modifications" part should be revised in the following sentence to clarify what this means in terms of the data presented.

CEP131 participates in optimal tubulin modifications

4- Ref 8 and 11 are the same.

5- Line 35, introduction: Instead of dismantling, centriolar satellite behavior during mitosis should be described as "dissolution" or "disassembly". Please also cite the following papers for mitotic dissolution of centriolar satellites: PMID 32286929 , 29973724

6- The following paper should also be included for introducing centriolar satellite phenotypes in the introduction. PMID: 32555591, 36790165, 33214552

Reviewer #2 (Remarks to the Author):

In this work, Renaud et al. studied the functions of CEP131, a protein interacting with the master organizer of the centrosome, namely scaffold protein PCM1 necessary for centrioles formation and homeostasis. CEP131 has been studied mainly in tumors and as complex in ciliated cells with PCM1. Not much is known about extra ciliary functions of these protein albeit present in non-ciliated cell. By KO CEP131 in a T-cell line they studied its involvement in cellular tubular arrangement as well as mitochondrial functions and cellular apoptosis.

In my opinion and with my knowledge, this work displays some intriguing findings and has the novelty of the regulation of some mitochondrial and tubular features by CEP131 in non-ciliated cells, which avoided confounding factors derived by ciliary signaling. Strength of the work is several methodological aspects and clear tubular signatures of the cells with KO of CEP131. The methods and the results as well as statistical analysis are well described and the figures are clear. I identified however some weaknesses which in my opinion need to be improved to give the right relevance and soundness to the discoveries. Moreover, the mechanisms of the data displayed are lacking. These will lift up the manuscripts significantly.

1. The authors use only one cell type which is a cell line and not even a primary cell type. This will be reiterated in my comments below. Some experiments will need to be addressed with other cell types especially primary to see if these are important mechanisms that work in different cells or they are specific of Jurkat cells. For several reasons described below and for the fact that are not ciliated cells relevant in biology, I suggest to perform some key experiments in primary hepatocytes or differentiated adipocytes (they also do not divide see comments on mitosis below)

2. Page 2 line 20 The authors perform KO in cells by selecting 2 different clones of transduced cell line. Are those deriving from a single cell? That might explain the differences in some aspects between these two clones such as mitochondrial membrane potential and other aspects Sup Fig1 KO1 vs KO2.

This is a point of concern since single cell integration by using lentivirus can be random and multicopy and affect several cell functions.

3. Page 2 line 30 Interesting finding of changing in tubular code by KO CEP131. The centrosome controls several microtubular cellular processes. Cellular division might be one of the most important and it is surprising that the authors did not study mitosis or cell cycle which might be impaired when metabolism or tubular organization is impaired. CEP131 overexpression has been shown to form abnormal satellite-like aggregates resembling abnormal centriolar satellites in autophagy deficient cells (Søs Grønbæk Holdgaard et al. 2019). This change mitosis and so the opposite might be true and needs to be investigated. CEP131 is not required for microtubule organization, but its KD decrease replicating cells and mitosis (Staples et al. 2012). However, in the present manuscript, the authors described that CEP131 participates in optimal tubulin modification and microtubule regrowth. Since only one cell line was used it might be very possible that some mechanisms are cell type specific (to address this see suggestions at point 1). The authors should address these discrepancies and find out if it is cell type specific.

4. Page 3 line 45 and mitochondrial experimental data Fig 3b.

A. It seems that the 2 generated KO cell lines display quite different membrane potential. It is just matter of increasing slightly the sample size and the membrane potential will show significant differences also vs the control. Seahorse analysis therefore should be carried out with both lines to find out if there is differences also in cellular respiration.

B. Seahorse data are not normalized (cell number) therefore only relative measurements can be taken in consideration for analysis. This includes the spare respiratory capacity but in percentage respective to basal as well as coupling efficiency. Since metabolic differences and mitochondrial network might be also due to different cell growth, the normalization on cell number would need to be present. Why the authors did not include ECAR measurements which will allow a better understanding on cellular metabolism after CEP131KO? Method section: OCR means oxygen consumption rate and not Oxygen rate concentration.

C. Mitotracker accumulation depends on membrane potential. Since this is somehow different, at least as trend, it would be good to calculate mitochondrial mass by qPCR with mtDNA/nDNA.

D. How did the author calculate the mitochondrial volume is not well explained and the picture of mitochondrial network (TOM20) seem to represent a single plane which can be imaged in different z position and give false feeling of the networks in a single cells. To analyze the network the full volumetric network (3D reconstruction) needs to be addressed in several cells in the preparation and 3 biological replicates. It can be imaged with mitotrackers with confocal microscopy not requiring superresolution.

E. Page 4 line 6 The authors stated: "Altogether, these data indicate that CEP131 participates in the mitochondria fitness by controlling fission/fusion dynamic and respiration." In my opinion the data shown do not allow this conclusion. Mitochondrial fitness is also a word that is difficult to understand what it means.

F. What are the proposed mechanism for mitochondrial changes upon CEP131 KO? Can it be due to random clone selection? Cell replication speed can affect mitochondrial network as well as all cell metabolism and thus can be a consequence of that? The authors should address experimentally these concerns using another cell type (see point 1).

G. What are the mechanisms by which CEP11 regulates apoptosis through the mitochondria?

5. Page 4 line 35 The authors stated: From a functional standpoint, we found that cell death was likewise hindered in CEP131 knockout cells exposed to stimuli that involved the mitochondria, such as Raptinal, inhibitors of the BCL-2 family members (ABT-199, S63845, A1155463), staurosporine, etoposide, and TNFa (Fig. 4g and Supplementary Fig. 2f). This is a very interesting observation. However no mechanisms have been investigated nor proposed.

6. PCM1 and CEP131 are substrates for autophagy. Do they need to be interacting to be used as substrates for autophagy? Is PCM1 in absence of CEP131 differently used as substrate?

7. Page 4 lane 23 The authors suggest that CEP131 KO is reducing apoptosis. However in subsequent

experiment they are suggesting that CASP3 cleaves CEP131. All the process is quite confusing and it is difficult to understand. It should be explained better and with timeline of events. My concern is if CEP131 affect apoptosis, how at the same time CASP3 or apoptotic processes in general affect CEP131. Which is the first and pivotal player in this interaction?

8. Following the previous comment. Page 5 line 20 the authors write "These data suggest that the impeded release of cytochrome c from mitochondria and subsequent cell death observed in CEP131 knockout cells results from intrinsic changes in mitochondria". No mechanisms are provided for this event which is a bit difficult to tie to the KO of CEP131 since it seems not to play a role in mitochondria themselves. Further, it seems that the KO cells are dying more? Was it not protected from apoptosis? With less CASP activity?

9. The authors conclude: "Taken together, our data provide insights into the spectrum of functions controlled by CEP131 and may offer opportunities to modulate the equipoise between life and death decisions." This conclusion if supported by the suggestions above, might hold true but only in this specific cell line. More evidence should be needed to convince the reviewer that there are general functions of CEP131 and not limited to a cell line used in the laboratory.

Reviewer #3 (Remarks to the Author):

The present manuscript by Renaud and colleagues studies the centrosomal protein CEP131 in T-cell line unable to form primary cilia. The authors describe a novel role for the protein CEP131 in regulating mitochondrial biogenesis. Cep131 has been mainly associated with cilium formation, but little information is known about its function in non-ciliary cells.

Collectively, authors suggest that in cells depleted of CEP131 the mitochondria metabolic function improves and a delay occurs in the mitochondria-mediated caspase activation pathway and apoptosis. This is an interesting study examining the potential links between CEP131 and mitochondria fitness. The manuscript is written well and presented clearly.

My concerns are listed below:

1. The methods are well explained except for the Seahorse experiment. The authors should give more information about cell seeding since Jurkat cells are non-adherent. Did the authors used any strategy for cells to adhere?

2. The authors also refer in the methods section that HeLa cells were used, but in the legend of the figures there is no reference to that line.

2. On page 3 the authors refer that "Because fusion of the mitochondrial network has been linked to higher respiration and ATP production".

This comment should be reviewed since mitochondria become hyperfused and elongated in senescent cells.

3. Depletion of Cep131 is known to inhibit cell proliferation and induce centriole amplification, chromosomal instability, and post-mitotic DNA damage. Did the authors assessed any of these parameters? Or performed a viability test for the KO cells.

4. Another concern about the CEP131 knockout cells is the autophagy commitment.

Mitophagy is one of the most important mitochondrial control mechanism that shares the autophagic machinery. Can mitophagy impairment contribute to the mitochondrial elongation found in these cells?

5. In Figure 1J, Figure 4, and supplementary figures the loading control for the western blots (GAPDH or Tubulin) is not stable. This should be reviewed since it may indicate protein loading variation.

6. In Figure 3 the authors should include the spare respiratory capacity of the cells. In the WT cells seem to be negative indicating no mitochondrial reserve or that the cells are already depolarized.

7. In the legend of figure 4 appears "(a) A densitometric analysis was performed and the ratio between Raptinal and DMSO conditions is shown (b). All values were then normalized by WT value (mean \pm SEM, n=3 biological replicates, one-way ANOVA; *p<0.05, **p<0.01)". First, a ratio between treated and CTRs is mentioned and then the values are normalized to WT. Can the authors please explain how the calculations were performed?

8. In Figures 3 and 4 MFI appears in the vertical axis label. The abbreviation should be included in the figure legend.

9. For the statistical analysis the authors used one-way and two-way ANOVA but did not specify which post-hoc tests were applied.

10. In my point of view the results are poorly discussed in the discussion section.

Soap
Signaling in Oncogenesis
Angiogenesis & Permeability

Nicolas Bidère
CIRCINA INSERM U1307/CNRS 6075/UN
SOAP: Signaling in Oncogenesis, Angiogenesis, and Permeability
8 Quai Moncoussu BP 70721
44007 Nantes Cedex 1
France
Ph: 33 68 352 6986
Email: nicolas.bidere@inserm.fr

Re: COMMSBIO-23-1615

Thank you very much for considering our manuscript for publication in *Communications Biology*. We wish also to thank the referees for their positive and insightful comments to enhance our manuscript. We have now had time to carry out substantial additional experimental work to address the queries of the reviewers. As you pointed out, we also acknowledged in the discussion the limitation of using mostly a single cell type.

Please find below our point-by-point discussion in which we reiterate the reviewers' points and provide a rebuttal.

Reviewer #1

1- CEP131 localizes both to the centrosome and centriolar satellites. In some parts of the manuscript, the authors concluded or discussed the phenotypes of CEP131 as centriolar satellite phenotypes, which should be revised as the phenotypes might also be linked to centrosomal pools of CEP131. It will be misleading to assign the new CEP131 phenotypes as centriolar satellite-specific phenotypes except for CASP activation and delayed apoptosis resulting from depletion of centriolar satellite marker PCM1 in Fig. S3.

We agree with the Reviewer and revised our manuscript to emphasize that CEP131 is not exclusively part of centriolar satellites. Indeed, CEP131 has been initially identified as a centrosomal protein by proteomic analyses and immunofluorescence in overexpression conditions ¹. It was subsequently shown that endogenous CEP131 also localizes to the core centriolar region and centriolar satellites in interphase ²⁻⁶. An additional layer of intricacy emerges from the dynamic and versatile positioning of CEP131 depending on the cellular context ⁷. For instance, CEP131 accumulates at the centrosome of mitotic cells and cells lacking centriolar satellites ^{2,3}. Moreover, cellular stresses such as UV radiations drive the MK2-mediated phosphorylation of CEP131 and its subsequent sequestration in the cytosol ^{8,9}. These interesting aspects are now included both in the introduction (page 2, lines 5-12) and discussion (page 6, lines 34-43) sections.

2- To determine the functional relationship between centriolar satellites and CEP131, CASP activation and apoptosis assays performed in CEP131 Kos or PCM1 depleted cells should be performed in cells depleted/deleted for both proteins. Comparative analysis of reported phenotypes with that of CEP131 KO depleted for PCM1 will allow authors to assess how loss of centriolar satellites (PCM1 depletion) changes CEP131-linked phenotypes.

This is an interesting suggestion, which we tried to address experimentally. Our results show that the defect in CASP3 activation resulting from CEP131 deletion is not drastically increased when PCM1 is silenced in Jurkat cells (new Supplementary Fig. 4d). Hence, PCM1 silencing somewhat mimics

CEP131 deletion but does not further enhance the observed phenotype. In the discussion section, we now speculate on the contribution of centriolar satellites in maintaining the proper spatial distribution of CEP131 and/or its cellular abundance (page 6, lines 34-43).

3- Along the lines of CEP131 having centrosome and centriolar satellite pools, the title should be revised to better represent the paper. Cep131 localizes to both centrosome and centriolar satellites, so this should be included. "timely" can be removed from the title.

In light of Reviewer#1 points #1 and #3, the title has been changed to "The Centrosomal Protein 131 participates in the Regulation of mitochondrial Apoptosis"

4- Can authors compare the results from western blot analysis of Cep131 KO lines in Fig. 1 with that of global proteomics in Fig. 2? This will be a good way for validating the proteomics data. Moreover, densitometry analysis for Fig. 1A is required to quantitatively report the changes in the cellular abundance of the centriolar satellite proteins in CEP131 KO cells. Although CCDC66 levels were concluded to not change, the blots do not represent this conclusion.

This is a very insightful suggestion. We now included a comparison of data from the proteomic with a densitometric analysis of our western blot analyses presented in Fig. 1. The results, which validate the proteomic analysis, are presented in the new Fig. 2b.

5- Characterization of CEP131 KO lines for potential changes in PTMs in Fig. 1F was only performed by western blot analysis with antibodies against different PTMs, which is not sufficient to conclude that CEP131 functions in this process. The emphasis on this data throughout the manuscript should be tuned down. This data can also be moved to the supplemental figures. Similar comments also apply to autophagy function in Fig. 1j.

Following the Reviewer's comment, we tuned down the results related to the tubulin code and proposed additional hypotheses in our revised manuscript. Moreover, this set of data is now in Fig. S1. Nevertheless, in light of the comments from Reviewer#2 (please, see his/her points#4F, 5, 9, 10) and Reviewer#3 (please, see his/her point#10), these data were used to speculate on putative molecular mechanism in the discussion section (page 6, lines 6-13). Prompted by the Reviewer and to improve clarity, the data related to CEP131 and autophagy were discarded from the revised manuscript.

6- Densitometry analysis is required for Fig. 4a as some conclusions for this figure do not match with the representative data.

We concur with the Reviewer's observation that interpreting this set of data could prove challenging. Although we repeatedly observed reduced recruitment of VDAC to a low-density fraction upon cell death in CEP131 KO cells, suggesting reduced mitochondria fragmentation, we recognize the potential for this data to yield speculative conclusions and introduce confusion. To enhance clarity, we have opted to omit this panel from the revised manuscript.

7- Supp. Fig. 4d includes only one cell micrography per condition to represent changes to CEP131 cellular distribution. This phenotype should be quantified like it was done in Fig. 1b,c.

The requested quantifications are now included in the new Fig. S5d.

8- The authors did not perform phenotypic rescue experiments with CEP131 to confirm the specificity of phenotypes. I understand that the characterization of two different CEP131 clones is chosen instead to show specificity. However, these clones were only validated by western blotting in Fig. 1. To show that these clones do not have the same CEP131 mutation after genome editing, sequencing of the clones is required, which will be important to show specificity.

In the revised version of the manuscript, we now provide a more comprehensive analysis of the CEP131 KO clones used. The genomic DNA sequencing identified different bi-allelic mutations in the

CEP131 gene of clones #1 and #2 (Please, see new Fig. 1a). Moreover, a list of *in silico* predicted off-targets is included in Supplementary Table 1.

Minor comments:

1- Pg.1 Introduction: Satellites move in a motor-dependent manner (both kinesin and dynein). Please revise the following sentence: Centriolar satellites are small non-membranous proteinaceous particles of 70-100 nm that crawl along microtubules in a dynein-dependent manner and gravitate toward the centrosome

We thank the Reviewer and corrected this sentence.

2- Fig. 1g: One datapoint for gly. tubulin is very different than the rest two. To be conclusive, it might be worth increasing the number of experimental replicates.

We agree and increased the number of repeat experiments (n=6; Please, see the new Fig. S1f).

3- Abstract: "optimal tubulin modifications" part should be revised in the following sentence to clarify what this means in terms of the data presented.

This has been fixed.

4- Ref 8 and 11 are the same.

We thank the Reviewer for catching this mistake. This problem has been fixed in the revised version of the manuscript.

5- Line 35, introduction: Instead of dismantling, centriolar satellite behavior during mitosis should be described as "dissolution" or "disassembly". Please also cite the following papers for mitotic dissolution of centriolar satellites: PMID 32286929 , 29973724

We changed as suggested and cited these publications.

6- The following paper should also be included for introducing centriolar satellite phenotypes in the introduction.

PMID: 32555591, 36790165, 33214552

These papers are now quoted in the revised version of the manuscript.

Reviewer #2

1. I suggest to perform some key experiments in primary hepatocytes or differentiated adipocytes (they also do not divide see comments on mitosis below)

Prompted by the Reviewer, we carried out new lines of experiments to evaluate cell cycle, proliferation, and signs of chromosomal instability in Jurkat cells. Furthermore, we examined cell death in L929 fibroblasts transfected with siRNA targeting CEP131. Nevertheless, as suggested by the Editor and this Reviewer, we also acknowledged the limitation of essentially using a single cell type in our revised study (page 6, lines 6-8).

2. Page 2 line 20 The authors perform KO in cells by selecting 2 different clones of transduced cell line. Are those deriving from a single cell? That might explain the differences in some aspects between these two clones such as mitochondrial membrane potential and other aspects Sup Fig1 KO1 vs KO2. This is a point of concern since single cell integration by using lentivirus can be random and multicopy and affect several cell functions.

As inferred by Reviewer#2, these two clones were obtained after single-cell selection, and this is now clearly indicated in the revised manuscript (page 7, line 43 – page 8, line 10). Following the recommendation from Reviewer#1, we conducted an additional characterization analysis of these clones and showed that they carry different bi-allelic mutations (please, see his/her point#8).

Although some variability was observed between the two clones (please, see below), our results

support the idea that CEP131 participates in apoptosis signaling. Prompted by the Reviewer, some cell death-related assays were also carried out in L929 cells and HT-29 cells with siRNA.

3. Page 2 line 30 Interesting finding of changing in tubular code by KO CEP131. The centrosome controls several microtubular cellular processes. Cellular division might be one of the most important and it is surprising that the authors did not study mitosis or cell cycle which might be impaired when metabolism or tubular organization is impaired. CEP131 overexpression has been shown to form abnormal satellite-like aggregates resembling abnormal centriolar satellites in autophagy deficient cells (Søs Grønbaek Holdgaard et al. 2019). This change mitosis and so the opposite might be true and needs to be investigated. CEP131 is not required for microtubule organization, but its KD decrease replicating cells and mitosis (Staples et al. 2012). However, in the present manuscript, the authors described that CEP131 participates in optimal tubulin modification and microtubule regrowth. Since only one cell line was used it might be very possible that some mechanisms are cell type specific (to address this see suggestions at point 1). The authors should address these discrepancies and find out if it is cell type specific.

Several of these insightful remarks are shared with Reviewer#3 (please, see his/her point#3). As pointed out, the ectopic expression of CEP131 was shown to drive mitotic abnormalities in U2OS cells (multipolar mitotic figures)¹⁰. Interestingly, HeLa cells silenced for CEP131 also display increased centrosome amplification, and increased rate of mitotic errors, in addition to signs of CIN^{2,5}. Centrosome amplification was however not observed in CEP131-silenced U2OS cells, suggesting variations depending on the cell type⁶. Moreover, the silencing of CEP131 in HeLa cells was shown to reduce the proportion of mitotic cells and slightly slow cell proliferation^{2,5}. Following the Reviewers' suggestions, several lines of experimentation were carried out in our experimental settings. First, the analysis of the cell cycle did not show drastic changes in CEP131 KO Jurkat cells although a decrease in cell proliferation was observed, supporting previous works^{2,5}. Similarly, the cell cycle was unchanged in L929 cells transfected with CEP131 siRNA (data not presented). Of note, we did not notice abnormal mitotic phenotypes in our cultures of CEP131 KO cells (data not shown). We further checked for hallmarks of chromosomal instability and DNA damage response and did not observe increased numbers of micronuclei in CEP131 KO cells. This was also true in cells treated with the MPS1 inhibitor Reversine, a compound known to induce micronuclei¹¹. Furthermore, the molecular markers of DNA damage and repair (P-Chk2 and P-ATM) remained undetectable in CEP131 knockout Jurkat. This set of additional experiments is now included in the new Fig. S1. Finally, following the Reviewer's recommendation, we also discuss the possible link between mitosis and mitochondrial dynamics in the discussion section (page 6, lines 26-32).

4. Page 3 line 45 and mitochondrial experimental data Fig 3b.

A. It seems that the 2 generated KO cell lines display quite different membrane potential. It is just matter of increasing slightly the sample size and the membrane potential will show significant differences also vs the control. Seahorse analysis therefore should be carried out with both lines to find out if there is differences also in cellular respiration.

Here, we would like to take this opportunity to thank the Reviewer for this remark, which prompted us to refine our initial observation. At the Reviewer's suggestion, we first developed a more standardized procedure with Mitotracker Green and TMRM and conducted additional repeat experiments. The results, presented in the new Fig. 3a validate that mitochondrial mass remains changed. This was further corroborated with qPCR experiments using primers specific for mtDNA and nDNA (please, see below and in Fig. 3b). However, as highlighted by the Referee, there was a slight but significant change in $\Delta\Psi_m$ in CEP131 KO cells (please, see the new Fig.3c). Of note, the Seahorse analysis of the second clone did not reveal an increase in respiration, suggesting clonal variability (new Fig. S3a). Hence, the observed defect in apoptosis observed without CEP131 is likely unrelated to cellular respiration. These data are included in the revised manuscript.

B. Seahorse data are not normalized (cell number) therefore only relative measurements can be taken in consideration for analysis. This includes the spare respiratory capacity but in percentage respective to basal as well as coupling efficiency. Since metabolic differences and mitochondrial network might be also due to different cell growth, the normalization on cell number would need to be present. Why the authors did not include ECAR measurements which will allow a better understanding on cellular metabolism after CEP131KO? Method section: OCR means oxygen consumption rate and not Oxygen rate concentration.

We apologize for not indicating this information in our initial submission and for this factual error. The Seahorse data were normalized (per 300,000 cells). This is now mentioned in the manuscript (page 12, lines 23-36).

C. Mitotracker accumulation depends on membrane potential. Since this is somehow different, at least as trend, it would be good to calculate mitochondrial mass by qPCR with mtDNA/nDNA.

MitoTracker green (MTG), in contrast to MitoTracker Red and Orange, accumulates in the mitochondrial matrix independently of mitochondrial membrane potential and is classically used to measure mitochondrial mass. Nevertheless, we further complemented our data with the suggested qPCR experiment with mtDNA and nDNA. This essentially shows no significant differences between control and CEP131 KO cells. This new data is now shown in the new Fig. 3b.

D. How did the author calculate the mitochondrial volume is not well explained and the picture of mitochondrial network (TOM20) seem to represent a single plane which can be imaged in different z position and give false feeling of the networks in a single cells. To analyze the network the full volumetric network (3D reconstruction) needs to be addressed in several cells in the preparation and 3 biological replicates. It can be imaged with mitotrackers with confocal microscopy not requiring superresolution.

We apologize for the lack of explanation. In the new Fig. 3, we now show max intensity projections of SIM micrographs. The volumetric analyses were measured in three independent cells/conditions and the experiment was repeated three times independently. We now provide a more comprehensive procedure explanation in the method section (page 11, lines 3-14), Figure 3, and in the Figure legend of the revised manuscript.

E. Page 4 line 6 The authors stated: "Altogether, these data indicate that CEP131 participates in the mitochondria fitness by controlling fission/fusion dynamic and respiration." In my opinion the data shown do not allow this conclusion. Mitochondrial fitness is also a word that is difficult to understand what it means.

We agree and edited this part.

F. What are the proposed mechanism for mitochondrial changes upon CEP131 KO? Can it be due to random clone selection? Cell replication speed can affect mitochondrial network as well as all cell metabolism and thus can be a consequence of that? The authors should address experimentally these concerns using another cell type (see point 1). G. What are the mechanisms by which CEP131 regulates apoptosis through the mitochondria?

As previously mentioned, we now show that CEP131 silencing with siRNA in L929 cells also reduces cell death induced by Raptinal, ruling out potential pitfalls. Nevertheless, how exactly CEP131 is linked to mitochondria and regulates apoptosis is intriguing, and will be the focus of our future work. In the discussion section, we posit that CEP131 depletion may alter the distribution of mitochondria in cells and change their ability to release cytochrome c, as was described for DRP1. We also raise the question of whether CEP131 changes mitochondrial dynamics by impacting discretely mitosis. It is important to note, however, that we have acknowledged this limitation in the discussion part of the manuscript (page 6, lines 6-32).

5. Page 4 line 35 The authors stated: From a functional standpoint, we found that cell death was likewise hindered in CEP131 knockout cells exposed to stimuli that involved the mitochondria, such as Raptinal, inhibitors of the BCL-2 family members (ABT-199, S63845, A1155463), staurosporine, etoposide, and TNF α (Fig. 4g and Supplementary Fig. 2f). This is a very interesting observation. However, no mechanisms have been investigated nor proposed.

As mentioned in the response to Reviewer#2, point #4F, we have included this limitation in the discussion part of the manuscript.

6. PCM1 and CEP131 are substrates for autophagy. Do they need to be interacting to be used as substrates for autophagy? Is PCM1 in absence of CEP131 differently used as substrate?

This is a very insightful remark. PCM1 binds GABARAP through an LIR motif and is, together with CEP131, a substrate for autophagy^{10,12}. For instance, starvation drives autophagic degradation of PCM1 and CEP131 in U2OS¹⁰. Accordingly, starvation with EBSS also led to the degradation of PCM1 in L929 cells and this was partly blocked by treatment with Bafilomycin A1. Yet, this was not overtly changed when CEP131 was silenced. Of note, PCM1 degradation was hardly seen in EBSS-treated Jurkat cells, similar to what has been previously observed in RPE-1 cells¹², suggesting cell-type specificities. Hence, these additional results suggest that the autophagic degradation of PCM1 occurs merely independently of CEP131. Of note, prompted by Reviewer#3, we also tested the impact of CEP131 on mitophagy and found no difference (Fig. 3g).

Analysis of PCM1 Abundance upon Autophagy Induction. L929 cells were transfected with control nonspecific (NS) siRNA or siRNA against CEP131. After 48h, cells were washed and incubated with EBSS for 4h. When indicated, Bafilomycin A1 (Baf A1, 200 nM) was added. **a**, Cell extracts were prepared and analyzed by immunoblotting with antibodies specific to the indicated proteins. **b**, Densitometric levels of experiments as in (a). Shown is the mean \pm SEM, n=3 biological replicates, two-way ANOVA; *p<0.05).

7. Page 4 lane 23 The authors suggest that CEP131 KO is reducing apoptosis. However in subsequent experiment they are suggesting that CASP3 cleaves CEP131. All the process is quite confusing and it is difficult to understand. It should be explained better and with timeline of events. My concern is if CEP131 affect apoptosis, how at the same time CASP3 or apoptotic processes in general affect CEP131. Which is the first and pivotal player in this interaction?

We apologize for not explaining better this aspect and for the resulting confusion. We conclusively show that CEP131 participates in the early steps of apoptosis by allowing optimal CASP activation. We also made the observation, that during the subsequent execution phase of apoptosis, when CASPs are fully processed, CASP3 cleaves CEP131 in addition to several centriolar satellite elements. The functional consequence of these many cleavages warrants further investigation. It could be a way to limit apoptosis or it could serve an additional role. Supporting this latter hypothesis, Seo and Rhee recently reported that the centrosomal proteins SAS-6 and pericentrin are cleaved by CASPs during apoptosis and that this may destabilize the centrosome organization and favor the formation of apoptotic microtubules¹³. This aspect is now discussed in the revised version of the manuscript (page 6, line 45 – page 7, line 5).

8. Following the previous comment. Page 5 line 20 the authors write "These data suggest that the impeded release of cytochrome c from mitochondria and subsequent cell death observed in CEP131 knockout cells results from intrinsic changes in mitochondria". No mechanisms are provided for this event which is a bit difficult to tie to the KO of CEP131 since it seems not to play a role in mitochondria themselves. Further, it seems that the KO cells are dying more? Was it not protected from apoptosis? With less CASP activity?

We apologize for the confusion here. Our data show that CASP activity and resulting cell death are delayed without CEP131. Moreover, less cytochrome c is released from mitochondria in cellulo or from mitochondria isolated from CEP131 KO cells, suggesting an intrinsic defect. Yet, the underlying mechanism remains unclear and this is now clearly mentioned and discussed in the revised version of the manuscript.

9. The authors conclude: "Taken together, our data provide insights into the spectrum of functions controlled by CEP131 and may offer opportunities to modulate the equipoise between life and death decisions." This conclusion if supported by the suggestions above, might hold true but only in this specific cell line. More evidence should be needed to convince the reviewer that there are general functions of CEP131 and not limited to a cell line used in the laboratory.

As pointed out earlier and following the suggestion from the Editor, we acknowledged the limitation of essentially using a single cell type in our revised manuscript. Nevertheless, prompted by the Reviewers, we now report that deleting or silencing CEP131 delays apoptosis in different cell lines and defining the underlying mechanism will be the focus of our future investigations.

Reviewer #3

1. The methods are well explained except for the Seahorse experiment. The authors should give more information about cell seeding since Jurkat cells are non-adherent. Did the authors use any strategy for cells to adhere?

We apologize for the lack of clarity. We now detailed this point in the Methods section of the revised manuscript (page 12, lines 23-36).

2. The authors also refer in the methods section that HeLa cells were used, but in the legend of the figures there is no reference to that line.

We thank the Reviewer and fixed this point.

2. On page 3 the authors refer that "Because fusion of the mitochondrial network has been linked to higher respiration and ATP production".

This comment should be reviewed since mitochondria become hyperfused and elongated in senescent cells.

The Reviewer is correct. This sentence has been edited.

3. Depletion of Cep131 is known to inhibit cell proliferation and induce centriole amplification, chromosomal instability, and post-mitotic DNA damage. Did the authors assess any of these parameters? Or performed a viability test for the KO cells.

This important point was also raised by Reviewer#2. Please, see our response to his/her point#3.

4. Another concern about the CEP131 knockout cells is the autophagy commitment.

Mitophagy is one of the most important mitochondrial control mechanisms that shares the autophagic machinery. Can mitophagy impairment contribute to the mitochondrial elongation found in these cells?

This is an interesting point we tried to address experimentally. Proper mitochondrial dynamics have been linked to mitophagy and vice versa. To test the impact of CEP131, Jurkat cells were treated

with the electron transport chain inhibitors oligomycin and antimycin, to depolarize mitochondria and thereby trigger mitophagy¹⁴. As expected, this led to the processing of OPA1, the accumulation of the kinase PINK1, which activates the subsequent ubiquitination of outer mitochondrial membrane substrate such as MFN1 and phosphorylates nearby ubiquitin moieties in control cells. Interestingly, the hallmarks of mitophagy were not changed in CEP131 KO cells. These results are now presented in the new Fig. 3g.

5. In Figure 1J, Figure 4, and supplementary figures the loading control for the western blots (GAPDH or Tubulin) is not stable. This should be reviewed since it may indicate protein loading variation.

At the Reviewer's request, a repeat experiment or an additional loading control is now shown for these blots.

6. In Figure 3 the authors should include the spare respiratory capacity of the cells. In the WT cells seem to be negative indicating no mitochondrial reserve or that the cells are already depolarized.

We initially did not include the spare capacity in the figure since the maximal respiration, which is the sum of the basal and the spare capacity, was indicated. In the revised version, both maximal respiration and mitochondrial spare capacity are indicated (new Fig. S3a). As expected, WT cells display low mitochondrial reserve.

7. In the legend of figure 4 appears "(a) A densitometric analysis was performed and the ratio between Raptinal and DMSO conditions is shown (b). All values were then normalized by WT value (mean \pm SEM, n=3 biological replicates, one-way ANOVA; *p<0.05, **p<0.01)". First, a ratio between treated and CTRs is mentioned and then the values are normalized to WT. Can the authors please explain how the calculations were performed?

We apologize for the lack of explanation. As per Reviewer#1 suggestion, this panel has been replaced.

8. In Figures 3 and 4 MFI appears in the vertical axis label. The abbreviation should be included in the figure legend.

We thank the Reviewer for catching this mistake and changed the legend accordingly.

9. For the statistical analysis the authors used one-way and two-way ANOVA but did not specify which post-hoc tests were applied.

This has been added in the Supplementary data 1.

10. In my point of view the results are poorly discussed in the discussion section.

The discussion section has now been extensively revised.

Please find attached the revised manuscript. We thank the Editor and Referees for their insightful comments, which have further clarified the manuscript, and we trust that it is now suitable for publication.

Sincerely,
N. Bidère

References

1. Andersen, J. S. *et al.* Proteomic characterization of the human centrosome by protein correlation profiling. *Nature* **426**, 570–574 (2003).
2. Staples, C. J. *et al.* The centriolar satellite protein Cep131 is important for genome stability. *Journal of Cell Science* jcs.104059 (2012) doi:10.1242/jcs.104059.
3. Wang, L., Lee, K., Malonis, R., Sanchez, I. & Dynlacht, B. D. Tethering of an E3 ligase by PCM1 regulates the abundance of centrosomal KIAA0586/Talpid3 and promotes ciliogenesis. *Elife* **5**, (2016).
4. Douanne, T. *et al.* CYLD Regulates Centriolar Satellites Proteostasis by Counteracting the E3 Ligase MIB1. *Cell Reports* **27**, 1657–1665.e4 (2019).
5. Denu, R. A. *et al.* Polo-like kinase 4 maintains centriolar satellite integrity by phosphorylation of centrosomal protein 131 (CEP131). *Journal of Biological Chemistry* **294**, 6531–6549 (2019).
6. Li, X. *et al.* USP9X regulates centrosome duplication and promotes breast carcinogenesis. *Nat Commun* **8**, 14866 (2017).
7. Renaud, C. C. N. & Bidère, N. Function of Centriolar Satellites and Regulation by Post-Translational Modifications. *Frontiers in Cell and Developmental Biology* **9**, 8 (2021).
8. Tollenaere, M. A. X. *et al.* p38- and MK2-dependent signalling promotes stress-induced centriolar satellite remodelling via 14-3-3-dependent sequestration of CEP131/AZI1. *Nat Commun* **6**, 10075 (2015).
9. Villumsen, B. H. *et al.* A new cellular stress response that triggers centriolar satellite reorganization and ciliogenesis. *EMBO J* **32**, 3029–3040 (2013).
10. Holdgaard, S. G. *et al.* Selective autophagy maintains centrosome integrity and accurate mitosis by turnover of centriolar satellites. *Nat Commun* **10**, 4176 (2019).
11. Toufekhtchan, E. & Maciejowski, J. Purification of micronuclei from cultured cells by flow cytometry. *STAR Protocols* **2**, 100378 (2021).
12. Joachim, J. *et al.* Centriolar Satellites Control GABARAP Ubiquitination and GABARAP-Mediated Autophagy. *Curr Biol* **27**, 2123–2136.e7 (2017).
13. Seo, M. Y. & Rhee, K. Caspase-mediated cleavage of the centrosomal proteins during apoptosis. *Cell Death Dis* **9**, 571 (2018).
14. Narendra, D., Tanaka, A., Suen, D.-F. & Youle, R. J. Parkin is recruited selectively to impaired mitochondria and promotes their autophagy. *J Cell Biol* **183**, 795–803 (2008).

Reviewers' comments:

Reviewer #1 (Remarks to the Author):

I would like to acknowledge the significant effort put forth by the authors in addressing the concerns and points I raised. It is evident from their revisions and responses to both my comments and those from other reviewers that they have conducted extensive and meaningful experimentation, which together significantly improved the manuscript. Specifically, they did a great job in addressing my concerns regarding the role of centriolar satellite and centrosome pools of Cep131 in the new functions they reported. I genuinely appreciate their diligence and dedication. Given the substantial improvements made, I recommend this manuscript for publication.

Reviewer #2 (Remarks to the Author):

After reviewing the manuscript several improvements have been made by the authors, which make the reader more aware of the results and the implications of the findings.

However, few points still need to be revised to make correct statements and report the correct data:

- Related to my previous point 3: the authors reported a cell cycle was unchanged in L929 cells transfected with CEP131 siRNA but they did not show the data. Same with the following statement on CEP131 KO cells. It should be shown.

- Related to my previous point 4 and B: at page 4 it is very important to note and describe that the membrane potential is statistically significantly increased in both clones as the data show. In fact, both cells have similar changes in mitochondrial network and it goes together with the similar change in membrane potential.

Moreover, the seahorse data are wrongly presented and incomplete. Related to my point B: The seahorse data are not normalized on cell number because this normalization is only possible after the run or the seeding. In fact, seeding the cells in the seahorse plate is known to cause quite a variable error. Therefore, only data on spare respiratory capacity and coupling efficiency can be shown and used to describe the cellular phenotype. Other data might generate confusion for the reader. To the values of basal, Maximal and oligomycin recordings needs to be subtracted the non-mitochondrial respiration. Only then the authors can show the basal, oligomycin and CCCP induced respiration in pmol/min/million cells (which is not possible in their case). However, the authors should show the spare respiratory capacity (max/basal OCR) or coupling efficiency $= (1 - (\text{oligo/basal})) * 100$. All these values need to be used after non-mitochondrial respiration. Showing the data in the way they presented might lead to incorrect evaluation of the cellular phenotype.

Reviewer #3 (Remarks to the Author):

The authors did a good job addressing all the issues raised.

There is only a small concern about the differences in mitochondria function (membrane potential and OCR) between the CEP131 KO #1 and #2 clones. How can the authors explain this?

Soap
Signaling in Oncogenesis
Angiogenesis & Permeability

Nicolas Bidère
CIRCINA INSERM U1307/CNRS 6075/UN
SOAP: Signaling in Oncogenesis, Angiogenesis, and Permeability
8 Quai Moncoussu BP 70721
44007 Nantes Cedex 1
France
Ph: 33 68 352 6986
Email: nicolas.bidere@inserm.fr

Re: COMMSBIO-23-1615A

Thank you for your interest in our work and the Reviewers' comments on our revised manuscript. We are delighted to see that Reviewers#1 and #3 enthusiastically endorse our manuscript for publication. Reviewer#2 also acknowledges that we addressed most of his/her initial points, but raises additional remaining concerns. Please find below a point-by-point disposition of the queries of the Referees with our answers.

Reviewer #1

I would like to acknowledge the significant effort put forth by the authors in addressing the concerns and points I raised. It is evident from their revisions and responses to both my comments and those from other reviewers that they have conducted extensive and meaningful experimentation, which together significantly improved the manuscript. Specifically, they did a great job in addressing my concerns regarding the role of centriolar satellite and centrosome pools of Cep131 in the new functions they reported. I genuinely appreciate their diligence and dedication. Given the substantial improvements made, I recommend this manuscript for publication.

We thank the Reviewer for enthusiastically endorsing our manuscript for publication.

Reviewer #2

After reviewing the manuscript several improvements have been made by the authors, which make the reader more aware of the results and the implications of the findings. However, few points still need to be revised to make correct statements and report the correct data:

We thank the Reviewer for his/her positive assessment of our work and the remaining comments.

1. Related to my previous point 3: the authors reported a cell cycle was unchanged in L929 cells transfected with CEP131 siRNA but they did not show the data. Same with the following statement on CEP131 KO cells. It should be shown.

At the Reviewer's request, this set of data is now shown in the supplementary Fig. 1c. We have chosen not to incorporate information regarding mitotic phenotypes into the revised manuscript, as further experiments are required to validate this initial observation. Nonetheless, we have acknowledged the significance of exploring this aspect in the discussion section.

2. Related to my previous point 4 and B: at page 4 it is very important to note and describe that the membrane potential is statistically significantly increased in both clones as the data show. In fact, both cells have similar changes in mitochondrial network and it goes together with the similar change in membrane potential.

We agree with the Reviewer and now clearly state that the membrane potential is significantly increased in both clones.

Moreover, the seahorse data are wrongly presented and incomplete. Related to my point B: The seahorse data are not normalized on cell number because this normalization is only possible after the run or the seeding. In fact, seeding the cells in the seahorse plate is known to cause quite a variable error. Therefore, only data on spare respiratory capacity and coupling efficiency can be shown and used to describe the cellular phenotype. Other data might generate confusion for the reader. To the values of basal, Maximal

and oligomycin recordings needs to be subtracted the non-mitochondrial respiration. Only then the authors can show the basal, oligomycin and CCCP induced respiration in pmol/min/million cells (which is not possible in their case). However, the authors should show the spare respiratory capacity (max/basal OCR) or coupling efficiency $= (1 - (\text{oligo}/\text{basal})) * 100$. All these values need to be used after non-mitochondrial respiration. Showing the data in the way they presented might lead to incorrect evaluation of the cellular phenotype.

We agree with the Reviewer that seeding adherent cells in the seahorse plate could introduce potential sources of errors. However, when seahorse analyses are conducted with non-adherent cells, we directly plate the cells on cell-tak-coated plates, and we consistently observe minimal variation in OCR measurements across different wells and independent experiments (Please, see the new Figure S3a). In light of the Reviewer's recommendation, we subtracted the non-mitochondrial respiration to Basal, Maximal, and oligomycin OCR values. The coupling efficiency of a cell can be defined as the proportion of mitochondrial respiratory rate used to drive ATP synthesis and is calculated as ATP production rate divided by the Basal respiration rate multiplied by 100, as described by the manufacturer (Agilent). This information was added to the Methods section. To provide a more organized and comprehensive presentation of the results, we have restructured the data as follows:

- Figure S3a depicts the OCR over time
- Figure S3b provides calculations of basal OCR and maximal OCR.
- Figure S3c shows the calculation of the coupling efficiency and spare capacity.

Reviewer #3

The authors did a good job addressing all the issues raised.

There is only a small concern about the differences in mitochondria function (membrane potential and OCR) between the CEP131 KO #1 and #2 clones. How can the authors explain this?

We thank the Reviewer for this positive assessment of our revised work. We concur with the Reviewer's observation that differences exist between the two clones concerning the OCR, for which we currently don't have a clear explanation. Performing additional unbiased proteomic and transcriptomic comparisons between these clones may shed light on this matter. In our present manuscript, we focused on the phenotype observed in both clones, and we acknowledged this apparent discrepancy.

We thank you for your hard work on our manuscript, and hope that it now complies with *Communications Biology* format. Feel free to contact me if questions arise or if I can be of any help.

Sincerely,
N. Bidère

REVIEWERS' COMMENTS:

Reviewer #2 (Remarks to the Author):

I would like to thank the authors to address in a very proper way the concerns raised and I would recommend this work to be published.

Soap
Signaling in Oncogenesis
Angiogenesis & Permeability

Nicolas Bidère
CICI2NA INSERM U1307/CNRS 6075/UN
SOAP: Signaling in Oncogenesis, Angiogenesis, and Permeability
8 Quai Moncoussu BP 70721
44007 Nantes Cedex 1
France
Ph: 33 68 352 6986
Email: nicolas.bidere@inserm.fr

Re: COMMSBIO-23-1615AIP

We are delighted to see that all Reviewers enthusiastically endorse our manuscript for publication.

Accompanying this letter is our final version of the manuscript together with the set of figures and supplementary figures.

Feel free to contact me if questions arise or if I can be of any help.

With best wishes,

Nicolas Bidère